# Energy Justice in Slum Rehabilitation Housing: An Empirical Exploration of Built Environment Effects on Socio-Cultural Energy Demand

**Ramit Debnath [1,2,*]** 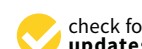 **, Gianna Monteiro Farias Simoes [3], Ronita Bardhan [1]** , **Solange Maria Leder [3]** , **Roberto Lamberts [4] and Minna Sunikka-Blank [1]**

1 Behaviour and Building Performance Group, The Martin Centre for Architectural and Urban Studies, Department of Architecture, University of Cambridge, Cambridge CB2 1PX, UK; rb867@cam.ac.uk (R.B.); mms45@cam.ac.uk (M.S.-B.)
2 Energy Policy Research Group, Judge Business School, University of Cambridge, Cambridge CB2 1AG, UK
3 Department of Architecture and Urbanism, Federal University of Paraíba, Joao Pessoa PB 58051-900, Brazil; gianna_farias@hotmail.com (G.M.F.S.); solangeleder@yahoo.com.br (S.M.L.)
4 Laboratory of Energy Efficiency in Buildings, Department of Civil Engineering, Federal University of Santa Catarina, Florianópolis, SC 88040-900, Brazil; roberto.lamberts@ufsc.br
* Correspondence: rd545@cam.ac.uk

**Abstract:** The interaction of energy and buildings institutes a complex socio-technical system that influences the eudemonic well-being of the occupants. Understanding these drivers become even more necessary in impoverished areas where occupants struggle to avail essential energy services. The literature indicates that energy injustice can be addressed through provisioning of comfort, cleanliness, and convenience (3Cs) as critical cultural energy services in low-income areas. This study investigates the socio-architectural influence for slum rehabilitation housing (SRH) on cultural energy services that can promote distributive justice. The methodology adopts an empirical route using data from 200 household surveys from SRH in Mumbai, India, and João Pessoa, Brazil. A model between the 3Cs and socio-architectural elements was established using Firth's binary logistic regression. The survey results showed that the SRH in Brazil had twice the appliance ownership as compared to the Mumbai SRH. There were distinct energy service preferences in the study areas, despite common poverty burdens. The empirical results showed that the lack of socio-architectural design elements like open spaces, privacy, and walkability in the study areas demanded specific comfort and convenience appliances as a counter-response. A critical policy implication drawn was on the need for socio-architectural inclusive energy planning for distributive justice in poverty. Mitigating rising energy demand through appropriate built environment design of slum rehabilitation housing can contribute to fulfilling the UN's SDG 7 (clean and affordable energy) and 11 (sustainable cities and communities) goals.

**Keywords:** poverty; energy justice; built environment; planning policy; slum rehabilitation; energy service; demand-side management; housing design

## 1. Introduction

An energy-just world is believed to promote happiness, welfare, freedom, equity, and due process for both producers and consumers [1] (p. 13). Energy justice is a critical element of contemporary energy policies addressing climate change mitigation and sustainable development goals. Energy justice frameworks have been designed to investigate and restructure the supply of energy and enhance equity [2]. Parallel to this approach, it is also essential to understand the human dimensions of energy

that determine the energy culture of a place [3]. Understanding energy culture can aid in designing "just" policies in a bottom-up targeted manner for equitable distribution of energy resources, especially for poverty alleviation [4,5].

Energy cultures are derived from everyday energy practices, norms, and the material reality of the built environment that drives the need for specific energy services [6]. An energy culture translates everyday energy consumption into household welfare, which promotes energy justice. It is the responsibility of an energy-just system to increase welfare by improving individuals' capabilities for maximising utility [7]. Distributive energy justice entitles people to a basic set of minimum energy services that enhance their eudemonic well-being [8]. However, the current literature lacks evidence on the thresholds of a minimum of energy services as energy consumption at a household level is principally viewed as a physical quantity that is measured in a standardised unit (kilowatt-hour (kWh)) [9].

Besides, at the individual level, energy is consumed in the form of "cultural energy services", which is driven by a complex socio-technical system of energy and built environment interaction [6,10,11]. This complex system is collectively referred to as human-scale energy services (HUSES) [12]. Anthropologist Elizabeth Shove (2003) [11] aptly describes the socio-technical forces behind socio-cultural energy services as comfort, cleanliness, and convenience (3Cs). "Comfort" is described as one's satisfaction with the immediate physical environment by controlling the built environment parameters of the indoor climate. "Cleanliness" is referred to as the energy services needed to maintain desired hygiene and sanitation conditions. It has a broader undertone of unique ideas of the display, disinfection, and deodorization of the built environment. "Convenience" refers to energy services that enable a smooth and effortless way of life. In the modern world, it is also associated with improving the quality of experience by using hyper-modern time-saving appliances (e.g., heating frozen food in microwave ovens rather than cooking every meal). The 3Cs are discussed in detail in Section 2.1.

Sovacool (2011) [13] applied the 3Cs concept of energy services to construct a theoretical urban energy service ladder that illustrated poorer-household demand energy services for subsistence. Middle-income households demand energy services for comfort, cleanliness, and convenience, whereas high-income households demand energy services for increasing consumption and convenience [13]. However, in reality, the urban poor exhibit a dichotomy in their consumption by portraying a middle-income consumption pattern [14,15]. We argue that this dichotomy is due to a cross-fertilisation of fulling aspirations of a middle-income consumption pattern and improving convenience through cultural energy services in poverty.

Owning a house is an aspirational element in the urban poor that shapes the cultural norms [16]. Slum rehabilitation aims at improving the quality of life and eudemonic well-being of the urban poor by enabling slum dwellers to own a house [17]. However, low-quality slum rehabilitation can negatively impact energy sustainability and health, well-being, and socialization of the urban poor [14,18,19]. A recent study on slum rehabilitation housing in India shows that a low-quality built environment pushes occupants towards energy poverty by increasing their household energy bills [20]. In the same study, the lack of open spaces has disrupted the social network of the occupants. We argue that the poor design of a slum rehabilitation built environment is a distributive injustice that is restricting the welfare benefits of cultural energy services (3Cs) in the study areas. Therefore, the influence of a slum rehabilitation built environment is investigated in the delivery of comfort, cleanliness, and convenience in poverty through appliance ownership.

This study's research focus is situated at the intersection of energy policy and built environment policy of hyper-dense cities of the Global South. It aims to solve the broader problem of identifying distributional benefits and costs of energy systems in rapidly urbanising cities under planning complexities. The novelty of this study lies in the empirical establishment of the socio-architectural needs and appliance ownership as critical 3C components that can be utilised for distributive justice-based policymaking. This study contributes significantly to the sparse literature on policy interaction for distributive justice from utility-side (electricity) and urban planning. Besides, it also contributes to the growing literature on the socio-technical understanding of architecture and energy

systems in rapidly urbanising cities of the Global South [21]. The cases of India and Brazil presented here aptly represent the complex urbanising scenarios where poverty alleviation efforts must be supported by just energy and climate policies [22]. Understanding energy consumption as 3Cs in poverty can help policymakers and utility companies in customising tariff mechanisms and ease the injustices due to the poverty trap. As the Global South prospers economically throughout this decade, millions of citizens will be moved out of extreme poverty through slum rehabilitation programs. It is therefore critical to improve its welfare effects, and a distributive energy justice perspective can guide future slum rehabilitation and energy sustainability policies.

To realise the research question presented above, we address the following objectives: a) To examine the variation in appliance ownership and energy practices in slum rehabilitation housing of Brazil and India as a description of cultural energy services; b) to investigate how socio-cultural energy services (comfort, cleanliness, and convenience) are derived through appliance ownership in the socio-architectural context of the study areas; and c) to empirically examine the role of socio-architectural variables of slum rehabilitation in the energy service demand for 3Cs through specific appliance ownership. A binary logistic regression is used to empirically answer Objective (c) using a 200-household sample survey on appliance ownership and socio-architectural amenities of the SRH in Brazil and India.

We vary two variables in this study—first, the typology of slum rehabilitation housing (SRH) (low-rise and high-rise buildings), and secondly the socio-cultural background of the occupants living in SRH (Brazilian SRH and Indian SRH) that defines their energy service needs. By varying these variables, we examine "how socio-architectural elements like access to open spaces, walkability, and comfort strategies influence the demand for comfort, cleanliness, and convenience (3Cs) through specific appliance ownership". We assume that the material manifestation of the 3Cs is through household appliance ownership, and just policies should enable low-income occupants to avail these services through appropriate socio-architectural design provisioning.

This study is structured as follows. Section 2 illustrates the applied energy concepts of cultural energy services and the literature evidence on the built environment—energy justice nexus. Section 3 presents the methodology with a detailed description of the study area and the study variables. Section 4 illustrates the results and contains the discussion, and has two subsections: Section 4.1 describes the exploratory results of energy culture in the study areas, and Section 4.2 illustrates the empirical result from Firth's binary logistic regression examining the influence of a lack of socio-architectural compatibility on cultural energy service demand. Finally, Section 5 presents the conclusion and policy implications of this study towards distributive energy justice.

## 2. Background

### *2.1. Cultural Energy Services (3Cs) and Appliance Ownership*

People do not consume energy in real life; they consume cultural energy services [10]. Such that the energy services can be specified through the conventions of comfort, cleanliness, and convenience (3Cs), which drive the energy consumption culture in society [11]. It is the cultural energy services that convert energy into well-being [8]. In a recent study, Brand-Correa et al. (2018) [12] explored the connection between well-being and energy use and called it human-scale energy services (HUSES). The authors found that household appliances act as critical transducers of energy to well-being conversion, such that HUSES are manifested through specific appliance ownerships. Here, we synthesise 3Cs as an applied energy concept (see Table 1) through the lens of the socio-cultural definitions of the 3Cs by Shove (2003) [11] and urban energy service ladder by Sovacool (2011) [13].

**Table 1.** Comfort, cleanliness, and convenience for appliance ownership as an applied energy concept.

| Cultural Energy Services | Shove's (2003) [11] Description of the 3Cs as Domains of Energy Consumption in Daily Life | Sovacool's (2011) [13] Interpretation of the 3Cs as Drivers of Urban Energy Consumption in Daily Life | Appliance Ownership as Material Manifestation of Energy Culture (3Cs) in the Slum Rehabilitation Housing (Authors' Assumption) |
|---|---|---|---|
| Comfort | A socio-technical system that co-evolved with the industrialisation of indoor climate and increasing energy intensity. It led to a worldwide standardisation of technologies, building styles and conventions, which now dictate the energy culture and the ownership of household appliances. For example: like owning cooling appliances, air conditioners, practices of opening/closing windows, etcetera. | The 3Cs are interpreted as one's satisfaction with the immediate physical environment, strongly associated with the ability to control indoor climate. It is a critical factor behind the global rise in air conditioning, especially among middle-income consumers. In low-income households, comfort has more economic connotations as income decides comfort outcomes. For example, poor households usually cater to natural ventilation, open spaces or fans for thermal comfort than energy-intensive mechanical cooling devices. | Thermal comfort: Fans and natural ventilation are the most common strategy [23]. Social comfort: Community areas and open spaces for socialising in the built environment. It is crucial for well-being [14]. Mental comfort: Community-feeling and preserving the social network in the built environment [14]. |
| Cleanliness | A co-evolutionary socio-technical and socio-cultural system that emerged from an identity-defining bathing and laundering practices to energy-intensive cleaning and laundering services. For example, change of bathing and hand-washing practices to a washing machine and hot-shower driven energy-intensive practices. It represents the change of cleanliness as a household practice to an industry-driven system of using detergents, washing machines, bathroom-fixtures, ironed-clothes, etcetera, for pleasure and duty. At a neighbourhood/societal scale, cleanliness-services is represented as the maintenance of hygiene and sanitation. | The social aspect of energy services that encompasses unique ideas of the display, disinfection, and deodorization. It also represents the energy need to maintain aesthetics, hygiene and sanitation in a household or a neighbourhood. | Electrification of cleaning regimes at a household-level. For example, aspirational uptake of washing machines as a "modern" device; vacuum cleaners and clothing irons. Better hygiene, safety, and sanitation in the built environment as a crucial need. |
| Convenience | Describe arrangements, devices, or services that helped save or shift time. A consumption culture where commodities and services are sold as being convenient or as making life more convenient for those who use them. There are modern and hypermodern forms of convenience devices that provide people with greater flexibility over their daily schedule that promotes "ease of life" or "welfare". For example, freezer, coffee maker, juicer, blender, smartphones, microwave ovens, computers/laptops/tablets, etcetera, are categorised as hypermodern devices. Convenience devices help in the branching of daily tasks through multi-tasking or in reducing the time of daily tasks that in turn, increases the demand for further convenience through the purchase of additional appliances. | Convenience can refer to reducing the effort needed to do a job as well as improving the quality of experience, such as watching a recorded show on a smartphone than on a tight television schedule. Lower-cost and enhanced services in today's age have put paramount importance on the "convenience" factor of owning an appliance. Services are needed round the clock and in an "instant". This demand for energy services is a primary reason for the rapid rise of energy demand in emerging economies like China and India, especially among middle-income consumers. | Uptake of hyper-modern appliances that saves time. It adds to the household welfare, especially to the women of the household by saving time from their daily chores [24]. For example, refrigerators, washing machine, microwave oven, coffee machine, mixer grinder, juicer, vacuum cleaners, etcetera Information and communication technology devices (ICT) like smartphones, TVs, Wi-Fi, laptops, computers, tablets, etcetera. |

Johnson, Gerber, and Muhoza (2019) [25] showed that the availability of energy services critically influences occupants' well-being. The demand for energy services is formed through specific energy practices, material culture, norms, and aspirations, which is met through household appliances ownership. It is referred to as the "energy culture" [6]. It establishes a logical link between appliance ownership and the demand for energy services, which cater to the socio-cultural need for comfort, cleanliness, and convenience (3Cs). Both Shove (2003) [11] and Sovacool (2011) [13] in their interpretation of social energy services converge their arguments on the welfare implications of appliance ownership, especially in fuel-poor and energy-poor households (see Table 1). Empirical studies from the Global South have also provided evidence on this association in low-income households. Like Dhanaraj, Mahambare, and Munjal (2018) [24] have found, welfare appliances like refrigerators and washing machines reduce the drudgery of women and children in doing activities like cooking, washing, and cleaning in low-income households of India. Reduction of drudgery saves time and improves convenience, which is, in turn, used for income generation contributing to household welfare [24]. Sovacool and Dworkin (2014) [1] in Chapter 7 provides its epistemology, which establishes the regimes of distributive energy justice through household welfare in poverty based on Amartya Sen and Martha Nussbaum's Capability Approach theory. We add another layer to this epistemology through the inclusion of socio-architectural design variables for just energy policymaking in poverty.

## 2.2. Built Environment and Its Influence on Cultural Energy Services

The literature on Social Practice Theory (SPT) has established critical theories between the material reality of the built environment and energy culture [11,26,27]. In SPT, individuals act as a carrier of a practice that ultimately leads to decision-making rather than just the behavioural attributes. Shove, Pantzar, and Watson's (2012) [28] elucidation on the material dimension (i.e., objects, infrastructure, tools, hardware, and the human body) of SPT established connecting theories around human–energy interactions in the built environment. Energy culture is one of the theories that connect energy practices with the socio-cultural norms and the material reality of the built environment [6].

A change in the built environment in low-income communities is linked with a change in their social processes. However, its energy implications are understudied. Identified drivers of residential electricity use include income, climate, demographic characteristics, energy price dynamics, dwelling type, and technology [29–31]. However, in the context of the Global South, energy studies have revealed a hierarchy in which appliances are acquired [32]. It is due to reliance on more than one energy source that causes a complex energy transition trajectory across the socio-economic domains. With the rise in household income, improved solutions become more accessible; there is a tendency to stack multiple energy sources, termed as "energy stacking" or "energy staircase" [33,34]. Energy stacking is common among the urban poor in the Global South, the reasons for which are not completely known yet. However, empirical studies have shown the influence of an energy culture on energy stacking practices, which point towards understanding the socio-cultural (i.e., non-income drivers) of energy demand in low-income communities [34–37]. We find that the above studies have investigated energy services as a socio-technical system that demands lighting, heating, cooling, entertainment, cooking, etcetera. It restricted the investigative boundary of the energy system as a physical quantity in these studies. This study expands this boundary by exploring the human-scale energy services that demand comfort, cleanliness, and convenience in the built environment; therefore, contributing to the growing literature on the non-income drivers of energy service demand and appliance ownership.

Besides, built environment quality, household size, automobile ownership, appliance characteristics, education level, gender dynamics, and household practices are also reported as critical non-income drivers of appliance ownership that drives energy consumption [29,38–40]. For example, Rao and Ummel (2017) [29] in their cross-country and micro-level study of Brazil, India, and South Africa have shown that penetration of appliances like television and refrigerators are highly driven by social practices, norms, and material culture across the social groups. Similarly, Debnath et al. (2019b) [20] have shown that slum rehabilitation in India influences high appliance

ownership due to the change in household practices. The change in the household practices is due to change in the built environment from a horizontal slum typology to a vertical rehabilitation housing typology. It indicates a possible influence of building typology in appliance uptake practices, which is investigated in this study.

In the rapidly urbanising Global South, the social impacts of the built environment and energy interaction in poverty were reported from Mexico. It was found that making energy efficiency retrofit in social housing would reduce the case of their abandonment by the overall improvement of occupants' "thermal comfort" [41]. Studies also showed that Brazil's infamous "My house, my life" national social housing program could have been made more effective and energy-efficient through an appropriate built environment design that can connect occupants to the community and improve their overall well-being. It involved providing access to open spaces, improving walkability, and setting up community terraces [42]. Similarly, bioclimatic design strategies at a neighbourhood level in Argentina's social-housing showed lowering of outdoor temperatures that, in turn, improved thermal comfort and reduced cooling energy demand [43]. In social housing of India, Bardhan, Debnath, Malik, and Sarkar (2018) [44] have shown that effective geometric and spatial arrangements of these housing units can improve the overall quality of life. The authors investigated the role of socio-architectural elements that improve indoor comfort and air quality. Thus, in low-income communities, built environment design acts as a critical catalyst in shaping the energy culture, which, in turn, determines the demand for cultural energy services.

## 2.3. Built Environment and Energy Justice: Intersection of Sustainable Urban Planning and Energy Systems

The built environment plays a critical role in realising the distributional benefits of energy justice-based policies through inclusive land-use and urban planning [45]. However, the distributional benefits of energy systems are often overlooked in urban planning narratives, as land-use zoning for infill and high-rise developments become a policy priority [45,46]. This approach has a similar effect in the planning of slum rehabilitation programs, where the aim of the developers remains to maximise occupancy and to fill the housing deficit [47]. In the slum rehabilitation housing of Mumbai, India, the high-rise development policy has severe negative ramifications on the quality of life of occupants as they get restricted fresh air, daylight, open spaces, sanitation, and hygiene conditions [18,19]. It accentuates some of the injustices of built environment planning that has a spillover effect in energy use and appliance ownership in such low-income houses [14,20].

Energy justice-driven urban planning is a contemporary topic in energy research and social science. A recent study showed that the current focus on urban planning and energy justice policies is in determining the spatial scope of the energy systems in cities. For example, Poruschi and Ambrey (2019) [45] investigated the spatial distribution of solar PV technology as a distributive energy justice measure to rising cooling and heating demand in Australian cities. They used a dynamic socio-economic panel data and geospatial dataset to determine the spatial location of solar PV panel installation for energy equity. Similarly, Byrne et al. (2016) [48] examined the potential of urban greening in reducing thermal inequality in Australia. The authors used a practice-based lens to understand occupants' disposition towards using green infrastructure to combat heat stress. In the US, Zhou and Noonan (2019) [49] used green building and smart meter roll-out programs across racially diverse neighbourhoods to investigate energy injustices. In doing so, the authors derived new energy justice exploration and policy perspectives [49]. From a city-planning standpoint, the concept of economic and social justice, along with energy sustainability, was used by Chatterton (2013) [50] to develop an agenda for post-carbon affordable communities in the UK.

Similarly, Sanchez and Reames (2019) [51] have used a socio-spatial analysis in justice-based policy design to mitigate urban heat islands using green roofs in Detroit, USA. It is one of the very few recent studies that have used energy justice as an urban planning philosophy. It indicated the need for justice-based pathways for addressing future cooling demands, especially in low-income and vulnerable communities [51].

Josa and Aguado (2019) [52] provided an in-depth review of cross-fertilising themes across economic, environmental, and social aspects in civil engineering, infrastructure planning, and society. They found that energy-justice in urban transportation and the mobility segment can have a broader social transformation effect at a city scale and derived a framework for holistic decision support for planners and policymakers. A critical methodological study was done by Heffron, McCauley and de Rubens (2018) [53] who developed an energy justice metric as a research and policy decision-making tool to tackle inequality. It used an environmental sub-parameter "Cost of Loss of Amenity to Local Communities" that connected the local built environment variables (amenity) with the direct and indirect effect of energy sources. The process of derivation of this energy justice metric provided a critical methodological clue to this study. Similarly, from an urban sprawl mitigation perspective, Wilson and Chakraborty (2013) [54] found that the current paradigm of planning research demands multidisciplinary considerations of resilience and environmental, energy, and climate justice for tackling urban informality and sprawl/slum formation. Our study expands on it by exploring the socio-architecture needs in poverty with the demand for specific energy services. This understanding can enhance justice-driven policymaking capabilities.

## 3. Materials and Methods

### 3.1. Data Collection and Survey Design

Data was collected in the slum rehabilitation housing (SRH) of Mumbai, India, and João Pessoa, Brazil—based on a comfort, cleanliness, and convenience (3C) appliance ownership survey. The survey questionnaire was designed based on the theoretical background of an energy culture to examine the socio-cultural factors influencing the demand for the 3Cs (after [6]). We specifically interviewed women of the household as they spend most of the time in their built environment. This occupancy pattern is distinct to low-income households in the Global South [55]. The classification of the appliances based on the 3C category is done as per Table 1. The survey variables are illustrated in Table 2. The data was collected in August 2019.

The questionnaire design and the surveys were conducted as per the best practice guidelines of UN-DESA (2005) [56]. The question frames were designed as per the energy culture categorisation, such that we can map the social process involved in the demand for 3Cs through energy practices, norms, and material culture in the surveyed households (see Supplementary Material S1 for detailed questionnaire). In doing so, we capture the time spent in the SRH during weekends and weekdays, thermal comfort perception at home as compared to living in the horizontal slums, what drives the use of cooling devices, along with the appliance ownership in the households. The surveys spanned across 200 housing units in Mumbai (n = 100) and João Pessoa (n = 100), which were selected using a stratified random sampling of the SRH units.

**Table 2.** Survey variables to explore energy culture in the slum rehabilitation built environments.

| Sl. No. | Survey Variables | Classification Category | Variable Type | Interconnected Energy-Culture Domains and Descriptions |
|---------|------------------|-------------------------|---------------|--------------------------------------------------------|
| A1 | Total appliance ownership | Energy use | Continuous | Material reality |
| A2 | Appliances owned | Energy use | Dichotomous (1 = Yes, 0 = No) | Norms and aspirations Comfort (thermal): Fan, air conditioners, air coolers, etcetera. Cleanliness: Vacuum cleaners, geysers, clothing irons, etcetera. Convenience: Welfare appliances: Washing machine and refrigerators Hyper-modern appliances: Microwave ovens, coffee machine, juicer, mixer-grinder, food mixer, DVD players, Smartphones, TVs, laptop, computer, tablets, etcetera. |
| A3 | Time spent at home in weekdays | Socio-cultural | Ordinal (1 = less than 12 h; 2 = 12–18 h; 3 = more than 18 h) | Practice |
| A4 | Time spent at home in weekends | Socio-cultural | Ordinal (1 = less than 12 h; 2 = 12–18 h; 3 = more than 18 h) | Practice |
| A5 | Thermal comfort perception at home as compared to horizontal slum | Built environment | Ordinal (1 = very cold; 2 = cold; 3 = slightly cold; 4 = neutral; 5 = slightly hot; 6 = hot; 7 = very hot; 8 = cannot answer; 9 = depending on the time) | Material reality |
| A6 | Strategies to improve thermal comfort | Socio-cultural and built environment | Dichotomous (1 = Yes, 0 = No) | Practice |
| A7 | Fan usage time at home | Socio-cultural | Ordinal (0 = Do not use/there is not, 1 = less than 12 h; 2 = 12–18 h; 3 = more than 18 h) | Practice |
| A8 | Window opening schedule (Day/Night) | Socio-cultural | Dichotomous (1 = Yes, 0 = No) | Practice |
| A9 | Reasons for keeping the windows closed | Built environment | Dichotomous (1 = Yes, 0 = No) | Material reality Lack of privacy; Risk of burglary; Entry of insect/dust; Used as storage; Noise; Rain; Solar gains; Broken windows |

### 3.2. Study Areas

#### 3.2.1. Mumbai, India: Slum Rehabilitation Housing (SRH)

The study area chosen in India is the slum rehabilitation houses in Mumbai in the state of Maharashtra. These houses are built under the "Slum Rehabilitation Housing" policy that redevelops slums into high-rise social housing by incentivising the private sector to participate in the redevelopment of slum communities. It provides legal entitlement to slum dwellers to a stipulated 25 m$^2$ apartment, including a bathroom with tap water and a kitchenette. In the past two decades, close to 0.15 million tenements have been rehabilitated using this model [57]. This policy provided the slum dweller access to a cross-subsidised, free-of-cost house, without burdening their time or economic poverty [47]. Recent studies have shown that these housing units lack the basic guidelines design, energy efficiency, or socio-cultural considerations [55] that imposes energy and health burdens on the occupants [18,57]. Households pay around 30–40% of their monthly income to electricity bills, making them vulnerable to energy poverty [20].

The specific survey location of SRH in this study is the Natwar Parekh Complex (NPC). The NPC, an SRH in the "M-ward" of Mumbai, was selected for this study. It is a high-rise SRH building with 15 floors and has 800 apartments (see Figure 1).

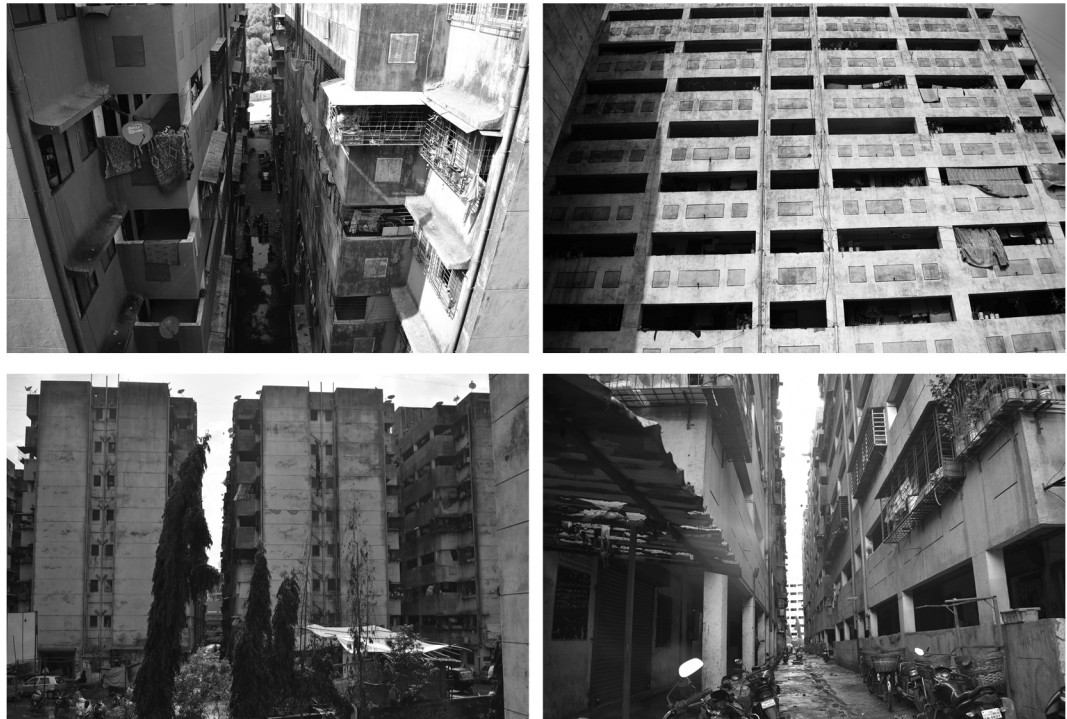

**Figure 1.** Slum rehabilitation housing of the Natwar Parekh Complex, Mumbai, India (source: authors).

#### 3.2.2. João Pessoa, Brazil: Gadanho and Timbó Social Housing (GTSH)

The study area in Brazil is in the city of João Pessoa, Paraíba State, northeast Brazil. Two social housing settlements were surveyed: Gadanho and Timbó, which were built in 2013 (see Figure 2). The Gadanho Social Housing has 45-row house units with one floor, whereas the Timbó Housing has two-storeyed houses with 136 units. These houses were built through a partnership between the City Council of João Pessoa and the Federal Government as a response to mitigate the housing deficit for the poor [58]. The GTSH rehabilitated slum dwellers who were affected by natural disasters in the city.. Each house in the GTSH scheme had a floor area of approximately 37 m$^2$, distributed across a living/dining room, one kitchen, one bathroom, and two bedrooms. Previous studies have shown that in most of the houses, there were post-occupancy refurbishments to maximise the living area [59].

Occupants added terraces in many houses through frugal construction methods that further impaired natural ventilation, leading to an increase in thermal discomfort. The GTSH built environment was designed with sidewalks and roads that improved walkability and access to communal spaces, which was an improvement as compared to the slums [59,60]. These houses were also built on the same neighbourhood where the resident lived previously, which makes it distinct from the slum rehabilitation houses in Mumbai.

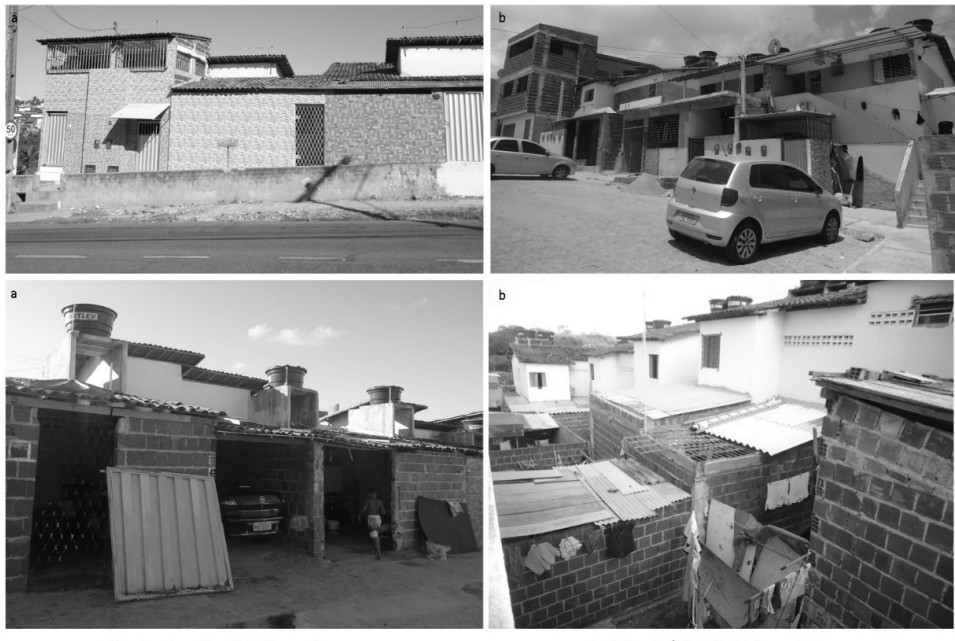

a. Gadanho Social Housing    b. Timbó Social Housing

**Figure 2.** (**a**) Gadanho Social Housing and (**b**) Timbo Social Housing in João Pessoa, Brazil India (source: authors).

However, existing literature also shows that the overall quality of these houses was of poor standards [59,60]. There was no consideration for thermal comfort and energy efficiency in the indoor design, making the GTSH uncomfortable [59]. As a compensatory measure, occupants perform frugal refurbishment of these dwelling units, which further deteriorates the indoor air quality by blocking natural ventilation [59,60]. Frequent refurbishments include adding a bedroom, increasing the kitchen size, or adding a terrace (like a veranda). These frugal refurbishments deteriorate the indoor thermal conditions and daylight conditions of the dwelling, decreasing or completely stopping the natural ventilation, as well as the daylighting [59]. The built environment and socio-economic characteristics of the surveyed households in both the study areas are illustrated in Table 3.

*3.3. Empirical Analysis: Cultural Energy Services and Essential Built Environment Design Element in Slum Rehabilitation Housing*

A binary logistic regression modelling approach was used to empirically estimate the influence of a lack of essential built environment design elements in the slum rehabilitation housing understudy with the demand for specific cultural energy services through specific appliance ownership. The essential built environment design variables that contributed to the reduction of distress and discomfort of the occupants in such low-income communities were adapted from Debnath et al. (2019a) [14]. We modelled five key variables concerning appropriate low-income built environmental design and planning, i.e., lack of privacy, lack of safety, walkability during daytime and night-time, and access to open/ventilated spaces in the neighbourhood (see Table 4). These dependent variables were collectively called as socio-architectural elements by [44]. Besides, the demand for comfort, cleanliness and convenience were empirically represented by the ownership of specific appliances, as per Table 1.

**Table 3.** Built environment, socio-economic, and energy-use characteristics of the study areas.

| Characteristics | SRH, Mumbai, India | GTSH, João Pessoa, Brazil |
|---|---|---|
| Building typology | High-rise (8 floors) | Low-rise (1–2 floors) |
| Built environment | Stacked buildings in a "shoe-box"-like manner. Poor provisioning of sidewalks and open spaces. Lack of hygiene and sanitation. Safety remains a problem. | Housing design was homogenous placed in an industrial manner. Well-defined sidewalks and roads. Safety remains a problem. |
| Floor area (m$^2$) | ~25 | ~37 |
| Spatial placement of rehabilitation houses | Away from slum location. | On the same location as the slums. |
| Average household income | USD 70–140 per month | USD 93.5–180 per month |
| Primary occupation of head of household (HoH) | Labourer in construction industry. | Labourer in waste-recycling industry. |
| Average number of people per household | ~5 | ~4 |
| Average education level of HoH | Middle-school | Middle-school |
| Average household electricity bill | USD 6–10 per month | USD 20–30 per month |
| Average household energy consumption (kWh) | 135 | 192 |
| Low-income electricity tariff program | None | None |
| Typical electricity demand drivers | Cooling (fans only), lighting, and entertainment | Cooling (fans only), lighting, leisure, and entertainment |
| Cooking fuel | Liquefied petroleum gas (LPG), kerosene | Liquefied petroleum gas (LPG) |

**Table 4.** Variable list for empirical modelling.

| Dependent Variable | Data Type (Binary: 1 = Yes, 0 = No) |
|---|---|
| Cultural energy service type in the study areas (E) (by specific appliance ownership) | (1) Comfort (ceiling fan, table fan, air-conditioners, air-coolers ownership) (2) Cleanliness (vacuum cleaners, geysers, clothing irons) (3) Convenience (Washing machine and refrigerators; microwave ovens, coffee machine, juicer, mixer-grinder, food mixer, DVD players; smartphones, TVs, laptop, computer, tablet) |
| **Independent Variable** | **Dummy Variable (Binary: 1 = Yes, 0 = No)** |
| Lack of socio-architectural built environment elements that are crucial for the well-being of occupants in slum rehabilitation housing. | (BE1) Privacy (BE2) Safety (BE3) Open space/ventilated space access during night-time (BE4) Walkability during daytime (BE5) Walkability during night-time |

The estimated value of specific cultural energy service demand (E, 1 = yes, 0 = no), was interpreted as the probability of the demand for comfort (E1), convenience (E2) and cleanliness (E3) (3Cs) in the respective slum rehabilitation housing neighbourhoods. The estimated model is illustrated in Equation (1):

$$E_i = b_0 + \beta_1 BE1 + \beta_2 BE2 + \beta_3 BE3 + \beta_4 BE4 + \beta_5 BE5 + u_i \qquad (1)$$

($E_i = 1$, if appliances for 3C were present; $E_i = 0$, if appliances for 3C were absent). Where $E_i$ indicated a binary variable corresponding to appliance ownership for specific cultural energy services, termed as comfort (Model 1), cleanliness (Model 2), and convenience (Model 3), respectively. Dummy variables were assigned (1 = Yes, 0 = No) for the dependent variables to match the above criteria of 3C-driven energy demand (see Table 4). Beta coefficients were represented through $\beta_1$ to $\beta_6$, and $u_i$ represented the error term of the model and $b_0$ was the intercept. Equation (1) tested the hypothesis of whether the lack of a specific socio-architectural design variable (BE1 to BE5, see Table 4) influences the energy service demand for the 3Cs.

Maximum likelihood (ML)-based binary logistic regression often fails to converge in a small sample [61]. The two most common concerns that arise from it are the loss of statistical power and bias and trustworthiness of standard errors and model fit tests [62]. Statistical power refers to the probability of finding significance when the alternative hypothesis is true in the population. It depends on the sample size, the variance of the independent and dependent variables, and effect size (e.g., odds ratio, proportional difference), among a few other things (e.g., number of predictors, the magnitude of the correlation among them, alpha level). For a detailed review of power and sample size estimation methods, refer to Bush (2015) [63].

ML estimation is known to have a small sample bias and produces an odds ratio that is too large for small samples [64]. Nemes et al.'s (2009) [64] estimation showed that the bias appears to be about 10–15% for the log odds ration when n = 100, and nearly entirely disappears as n = 1000. Thus, it was concluded that smaller samples could be expected to have a larger bias. Standard errors and significance tests require caution for smaller sample sizes in ML estimations (n < 100) [62]. The Wald test also performs poorly for small sample sizes [65]. To overcome these problems associated with a small sample size in ML estimates of binary logistic regression, Firth (1993) [66] introduced a penalised log-likelihood method. Firth's penalisation [66] has garnered significant attention as a method to reduce the small-sample bias of ML coefficients. Mathematically it can be represented as in [67].

Let $Y_i$, *(i = 1, 2, ... ,n)* be a binary outcome (0/1) for the *i*th subject, which follows a Bernoulli distribution with the probability $\pi_i = \Pr(Y_i = 1)$. The logistic regression model can be defined as Equation (2):

$$\text{Logit}[\pi_i|\boldsymbol{x_i}] = \eta_i = \boldsymbol{\varphi}^T \boldsymbol{x_i} \tag{2}$$

where $\boldsymbol{\varphi}^T$ is a vector of regression coefficients of length (k+1), and $\boldsymbol{x_i}$ is the *i*th row vector of the predictor matrix $\boldsymbol{x}$ which has order *n* x *(k+1)*. The term $\eta_i = \boldsymbol{\varphi}^T \boldsymbol{x_i}$ is called the risk score or "prognostic index". In standard ML, the model is fitted by maximising the log-likelihood denoted by *l(φ)*, whereas in penalised methods, *l(φ)* is maximised subject to constraints on the values of the regression coefficients. The penalised regression coefficient is obtained by maximising the penalised log-likelihood denoted by *l(φ)*–pen*(φ)*, where pen*(φ)* is the "penalty term". The penalty term is the functional form of constraints.

Firth [66] removed the first-order bias in the ML estimations of the regression coefficient by using the penalty term $\frac{1}{2} \, trace\left[I(\varphi)^{-1}\frac{\partial I(\varphi)}{\partial \varphi_j}\right]$ in the score equation $U(\varphi_j) = \frac{\partial I(\varphi)}{\partial \varphi_j} = 0$. The modified score equation is then represented as (see Equation (3)):

$$U(\varphi_j)^* = U(\varphi_j) + \frac{1}{2} \, trace\left[I(\varphi)^{-1}\frac{\partial I(\varphi)}{\partial \varphi_j}\right] = 0, \; j = 1, \, \ldots, \, k \tag{3}$$

where $I(\varphi)^{-1}$ is the inverse of information matrix evaluated at $\boldsymbol{\varphi}$. The corresponding penalized log-likelihood function for the above-modified score function is $l(\varphi) + \frac{1}{2}\log|I(\varphi)|$. It is known as Jeffreys invariant prior, and its influence is asymptotically negligible. The Firth type penalised ML estimator of $\boldsymbol{\varphi}$ is thus $\hat{\varphi} = argmax \left\{l(\varphi) + \frac{1}{2}\log|I(\varphi)|\right\}$. This method is bias preventive rather than corrective [67]. We fitted the binary logistic regression model (see Equation (1)) using Firth's bias reduction method, as illustrated above. It was proposed as the ideal solution to the problems of separation in logistic regression, especially with small samples [68]. The logistf package in R v3.3.3 was used for Firth's reduced-biased regression computations [69].

## 4. Results and Discussions

*4.1. Appliance Ownership and Energy Culture in Slum Rehabilitation Housing of João Pessoa, Brazil, and Mumbai, India*

Results show the distinction between the appliance ownership pattern in Joao Pessoa, Brazil, and Mumbai, India, households. Descriptive data shows that the total appliance ownership in Brazilian households is twice than that of the Indian households (see Figure 3). Welfare appliances like washing

machines and refrigerators occupy a significant portion of the total appliance ownership in both the case studies (see Figure 3). Welfare appliance ownership contributes to improved convenience in low-income households [24]. However, there are more refrigerators per household in the Brazilian case study (111/100) than in Indian households (61/100). This pattern continues in the washing machine ownership as well (Brazil (53/100); India (35/100)). Televisions (TVs) and fans are the most common household appliances in both the study areas. There are 152 fans in 100 surveyed households in Brazil, whereas in the Indian case there are 99 fans in 100 households. Most of the fans in the Indian case are ceiling fans, and the Brazilian households have both ceiling and table fans.

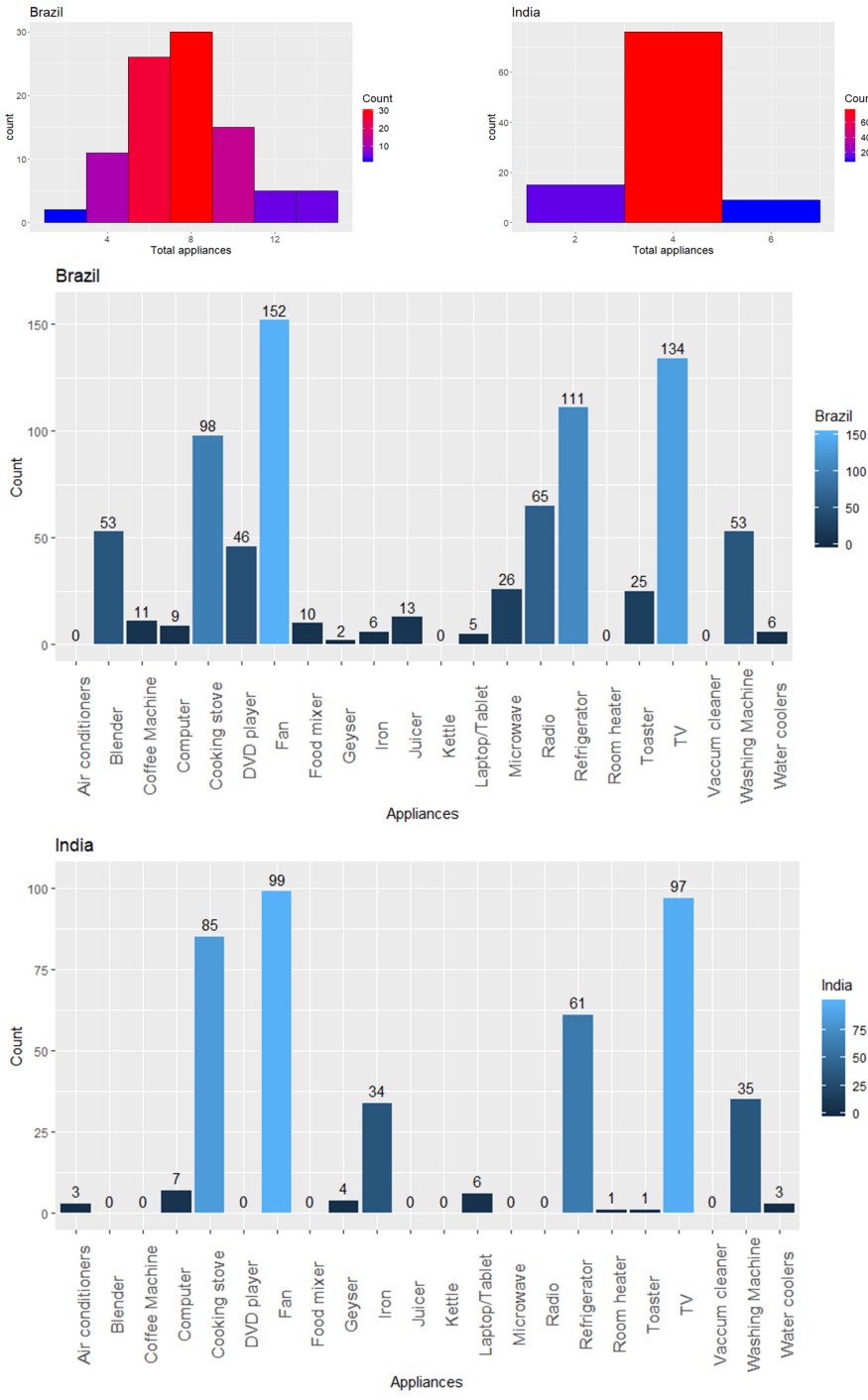

**Figure 3.** Household appliance in the slum rehabilitation housing of João Pessoa, Brazil (n = 100), and Mumbai, India (n = 100).

Similarly, for the TVs, Brazilian households had 132 TVs out of 100 samples, the Indian households had 97 TVs out of 100 samples (see Figure 3). Higher TV ownership can indicate more substantial demand for convenience-driven energy services in Brazilian households. Besides, higher ownership of hyper-modern appliances in GTSH may indicate a higher demand for convenience-related energy services (see Figure 3 and Table 2) [70]. However, higher appliance ownership in Brazil may be attributed to lower costs of appliances as compared to the Indian market [70]. It is beyond the scope of this paper to report such comparative energy market analysis results.

Higher ownership of fans indicates a comfort-based energy culture of mitigating thermal discomfort due to the hot and humid climate of both the study areas (see the Appendix A). The built environment design of both study areas is also reported to cause significant thermal discomfort (see Section 3.1). The exact reasons for higher TV ownership are not known; studies have shown that the popularity of "soap operas" act as cultural drivers of TV ownership in Brazil [71]. However, it indicates a media-consumption culture that is often interpreted as a compensatory response towards poor mental well-being [72,73]. In Mumbai, a recent study has shown that women feel lonely in slum rehabilitation housing (SRH), which motivates them to watch more TV and purchase more appliances as a compensatory mechanism [14].

Besides, survey results show that appliances like blenders, DVD players, coffee machines, juicers, microwave ovens, toasters, food mixers, and radios were exclusively present in the Brazilian case as compared to Indian survey households (see Figure 3). These appliances are categorised as hypermodern devices and are solely created for improving the convenience factor (see Table 2). Results also show that ownership of freezers, sewing machines, printers, air-fryers, video game consoles, bedside lamps, and home theatre systems in the Gadanho and Timbo Social Housing (GTSH) were absent in the SRH case (see Figure 3). Although the ownership of such hypermodern devices was low or even singular in some cases, it demonstrates the possibility of a more substantial convenience-driven energy culture in the GTSH as compared to the SRH. In the SRH, freshly ironed clothes have a significant social notion attached to it [20]. They are embedded deeply into the energy culture through higher ownership of clothing irons as compared to the GTSH case (see Figure 3).

The electrification of cleanliness was not clear from the survey results as both GTSH and SRH households performed manual cleaning of households; no vacuum cleaners were found (see Figure 3). There were no electric showers in both the survey areas, though the GTSH had more washing machines than the SRH, as illustrated in Figure 3. An electric shower is a standard appliance in middle-income households in Brazil and India. In both cases, washing machines were kept in either the kitchen or the living room due to a small floor area of the housing units, which creates severe space constraints. Occupants usually expand their rooms by frugal refurbishments in the GTSH to accommodate such appliances, which cause thermal discomfort and a lack of daylight. Simoes and Leder (2018) [59] reported that such refurbishments forced the occupants to buy additional fans and always use artificial lighting. It increased the overall energy intensity of these households. Such refurbishments are widely performed due to the low-rise building typology across Brazil [74]. However, such refurbishments were absent in the high-rise typology of the SRH. Still, these houses were affected with reduced daylighting and ventilation conditions due to poor design [14].

Occupancy pattern is a critical indicator of energy culture in the households. Figure 4 illustrates the occupancy pattern in the GTSH, Brazil, and the SRH, India, during the weekdays and weekends. In both areas, most of the surveyed occupants tend to stay indoors for more than 18 h a day, which is a noticeable characteristic of low-middle income housing [23]. This level of occupancy is due to the strong cultural norm that at least one member (mostly women) of the family stays at home to take care of children and the elderly (also reported in Bardhan and Debnath (2016) [75] for low-middle income housing in Mumbai). Such extended occupancy demands for cultural energy services concerning TV viewing and indoor energy-intensive practices [20,71]. For example, it can be seen in Figure 4 that both in Brazil and India, most of the surveyed occupants have an indoor occupancy of more than 18 h in the weekends and weekdays. Therefore, it is crucial to make indoor areas comfortable to balance

discomfort with energy-intensive cooling practices. A rise in indoor energy intensity is linked to higher indoor discomfort due to poor ventilation and thermal comfort levels in low-income households [76].

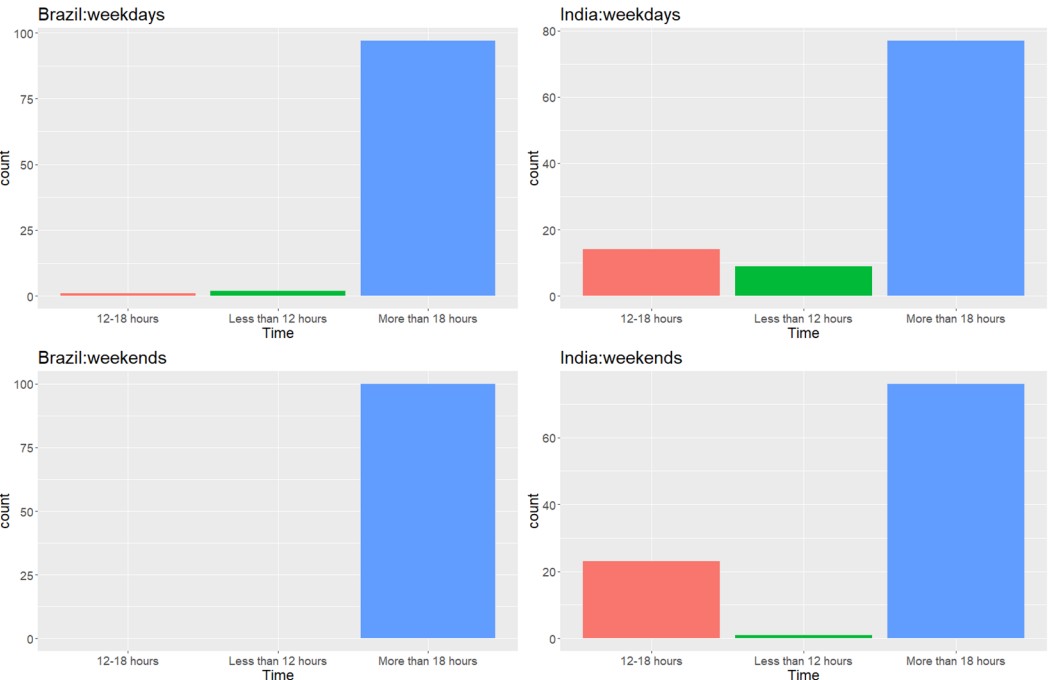

**Figure 4.** Time spent indoors in weekdays and weekends, depicting occupancy norms in the study areas.

The thermal comfort perception of the surveyed social housing in Mumbai and João Pessoa revealed high thermal discomfort (see Figure 5a). The GTSH occupants responded that their current homes as either "hot" or "very hot", as compared to "neutral" by the SRH occupants. The cooling device usage time is shown in Figure 5b, which shows ceiling fans as the most operated device in both India and Brazil. Climatologically, João Pessoa is less hot and humid than Mumbai (see the Appendix A). Physiologically, occupants of the SRH may have higher temperature tolerance than that of GTSH; it is beyond the scope of this study to investigate this aspect. However, as discussed above, the GTSH occupants perform extensive refurbishment of their low-rise housing units that block the windows, causing thermal discomfort [59]. To mitigate this discomfort, occupants in the GTSH used table fans in addition to ceiling fans as primary cooling devices (see Figure 5b). Thus, discomfort caused by frugal refurbishments of the built environment in GTSH is shaping the energy culture of high fan ownership.

Common strategies associated with maintaining the thermal comfort in GTSH and SRH is illustrated in Figure 6. It highlighted the energy culture associated with thermal comfort in the study areas. The most common thermal comfort practices in both the case studies were opening/closing of doors and windows and the use of fans (see Figure 6). Comfort measures that differed, like "taking a bath" and "adjusting clothing levels", were governed by the distinct socio-cultural norms of the study areas (see Figure 6). For example, bathing was not a standard thermal comfort response in the Indian case because it had a strong religious significance. Bathing as a practice in SRH households was done as a part of a religious routine, followed by wearing "freshly ironed clothes" (therefore, iron ownership was more in the SRH as compared to the GTSH, see Figure 3). However, in the Brazilian case, the survey showed that bathing was a direct response to thermal discomfort, so it stood out as a standard thermal comfort measure in the GTSH (see Figure 6).

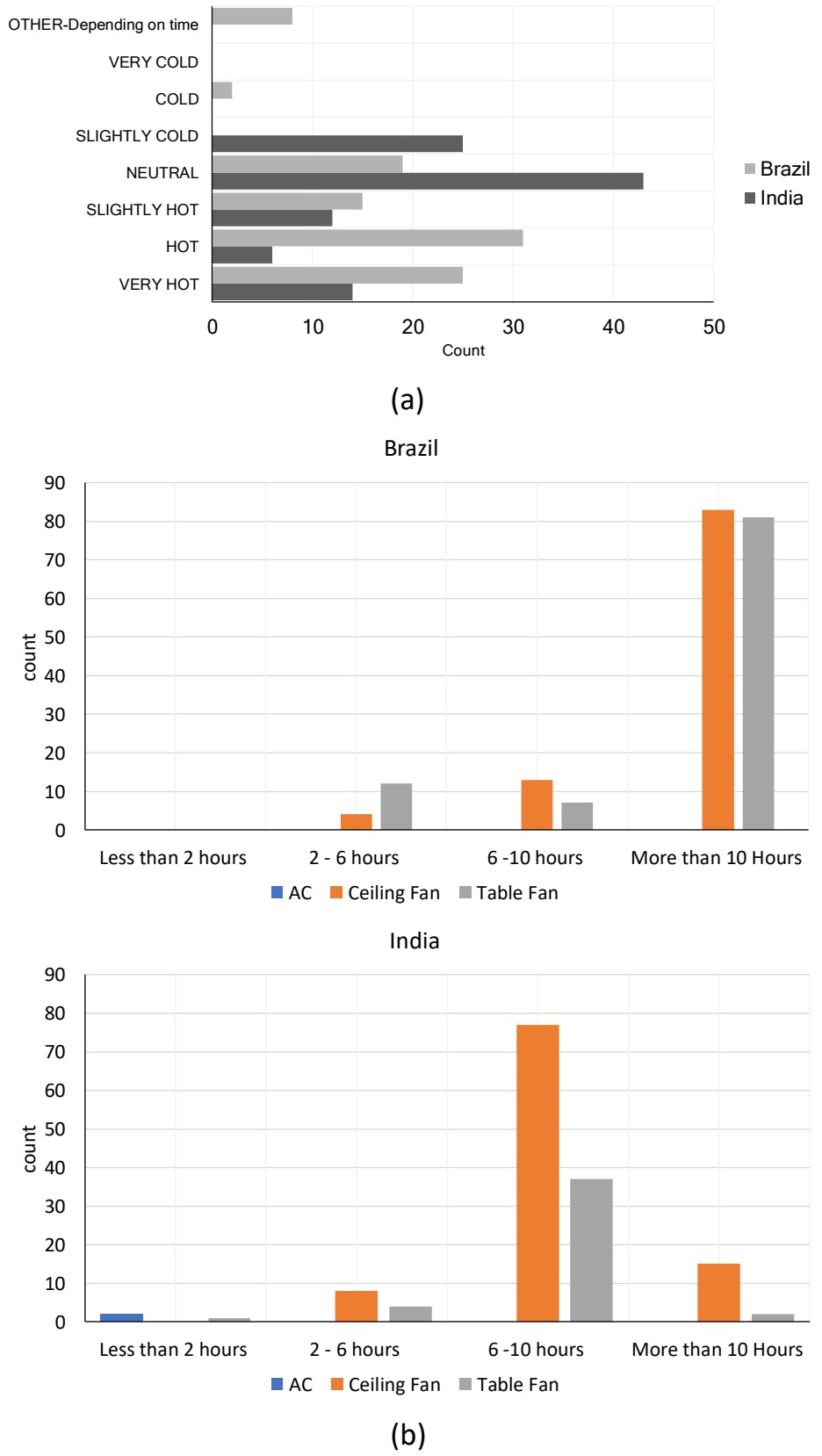

**Figure 5.** (**a**) Thermal comfort perception and (**b**) use of cooling devices (fans, table fans, and ACs) in the surveyed Brazilian (n = 100) and Indian (n = 100) households.

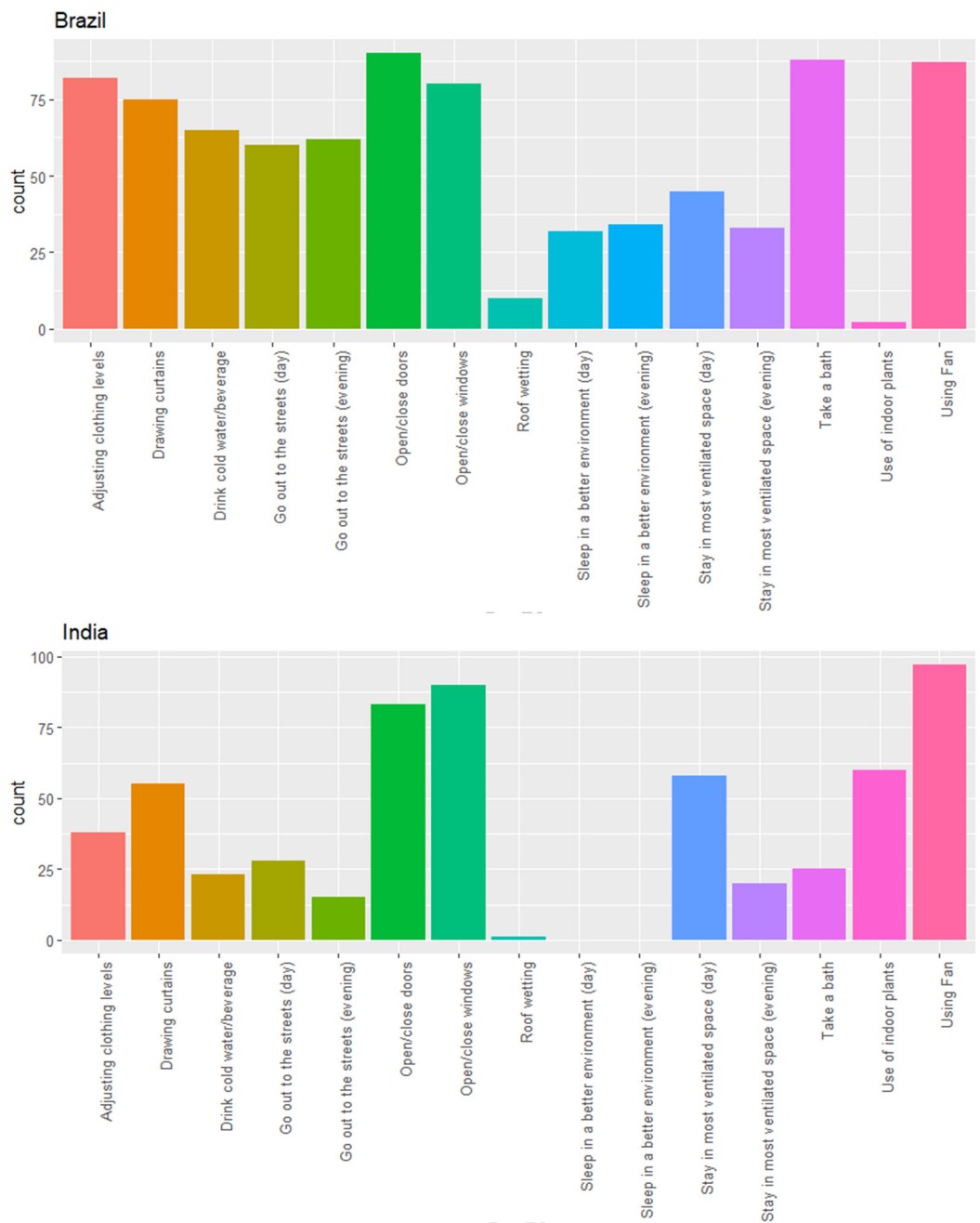

**Figure 6.** Practice and norms in GTSH, Brazil (n = 100), and SRH, India (n = 100), to restore thermal comfort in the built environment.

Similarly, the practice of adjusting clothing levels as a thermal comfort practice varied in SRH and GTSH, possibly due to the cultural norm of wearing distinct clothing styles. In the Indian case, "saree" was the most common women wear. It is a traditional piece of long cloth draped around the body. The clothing insulation (clo) values differ based on the time of the year: A typical winter ensemble of saree provides 1.10–1.39 clo, and summer and monsoon ensembles provide 0.62–0.96 clo. While the clothing adjustment values have high variability between summer and winter months, adjusting sarees to thermal discomfort was also governed by the degree of convenience [77]. On the contrary, Brazilian clothing norms were distinctively "western" ( clo-value varies between 0.5 to 0.7), which were more convenient to adjust to the thermal comfort [78]. Therefore, survey responses showed that "adjusting clothing levels" were more common in Brazilian households than the Indian households (see Figure 6).

Besides, Figure 6 also show the built environment-driven thermal comfort measures that were distinct to the socio-architectural characteristics of the study areas. Strategies like "going out to the street during the day and evening" were common in the Brazilian case as compared to the Indian case. It can be attributed to the low-rise typology of the GTSH, with walkable roads and dedicated community and open spaces (see Table 3). However, it does not mean GTSH has better roads and open spaces in terms of urban design. It is relatively better than the SRH context of Mumbai.

Walkability was reduced in the high-rise built environment of the SRH, which forced the occupants to remain confined in their housing units even during hot summer days. Besides, a lack of open and community spaces in the SRH also disturbed the community-cohesion of the occupants, and it affected their eudemonic well-being (also reported in [14]). Shifting of household practices to indoors and a lack of open spaces was found to be a significant reason behind the increase in the energy intensity of the occupants. It is leading the households in SRH to higher energy bills and posing greater vulnerability towards energy poverty [20].

The GTSH in Brazil was built on the same site as the slums. Owing to its low-rise typology, families feel more connected as their social network remains intact (also reported in [59]). It is one of the plausible reasons for the occupants to walk on the streets during day and night to mitigate thermal discomfort (see Figure 6). It provides further evidence on the influence of built environment design on the energy culture of households, which influences the comfort–convenience regimes of that place. Thus, space planning in slum rehabilitation housing is critical to the eudemonic well-being of the occupants. It indicates a planning-derived route to energy justice in such low-income communities.

Safety, hygiene, and sanitation of the built environment are critical planning components that are essential for the eudemonic well-being of the occupants [79]. These variables were often overlooked in slum rehabilitation housing planning that contributed significantly to occupants' distress and discomfort [14,18]. In doing so, we investigated the window operating schedules, as these were the only means of natural ventilation in both the study areas. Fresh air exchanges are critical in maintaining comfort, cleanliness, and convenience in low-income houses [44,79–81]. Figure 7 illustrates the windows opening and closing schedule in the study areas. Besides, it also demonstrates the built environment-governed reasons for keeping the window closed.

In both the slum rehabilitation housing, it was a common practice of keeping windows open during the day (see Figure 7). A few households were found to keep their windows open at night. It was surprising because, at night, all the family members were at home, which increased the occupant density of these housing units that increase the indoor temperature. It was a source of thermal discomfort in these housing units. It was the socio-cultural norms associated with privacy and the cleanliness component of the built environment that motivated window closure at night in the Indian case.

Windows were kept closed during the night due to high concentration of dust and insects in the SRH, Mumbai (see Figure 7). A high concentration of insects and dust were present due to lack of hygiene and sanitation (see Figure 1). Occupants threw garbage in the narrow space between the SRH buildings that posed a severe health and hygiene challenge (see Figure 1). Survey responses showed that the lack of open spaces and hygiene regulations in this built environment contributed to such practices. Similar observations were also reported by Kshetrimayum et al. (2020) [81].

On the contrary, survey results showed that the neighbourhood hygiene conditions were comparatively better in the Brazilian case. It was attributed to its low-rise building typology and regular access to social spaces (see Figure 2). Therefore, the influence of insect infestation and dust in keeping the windows closed in GTSH was small (see Figure 7). Lack of safety was a significant issue in both the study areas, such that occupants closed windows at night to prevent burglary (see Figure 7). These were some of the socio-architectural factors that influence the closure of windows at night when the occupant's density was the highest. The closure of windows at night caused thermal discomfort [14] that demanded energy-intensive cooling devices in both the study areas.

Such socio-architectural variables were empirically tested with demand for cleanliness, comfort, and convenience in Section 4.2.

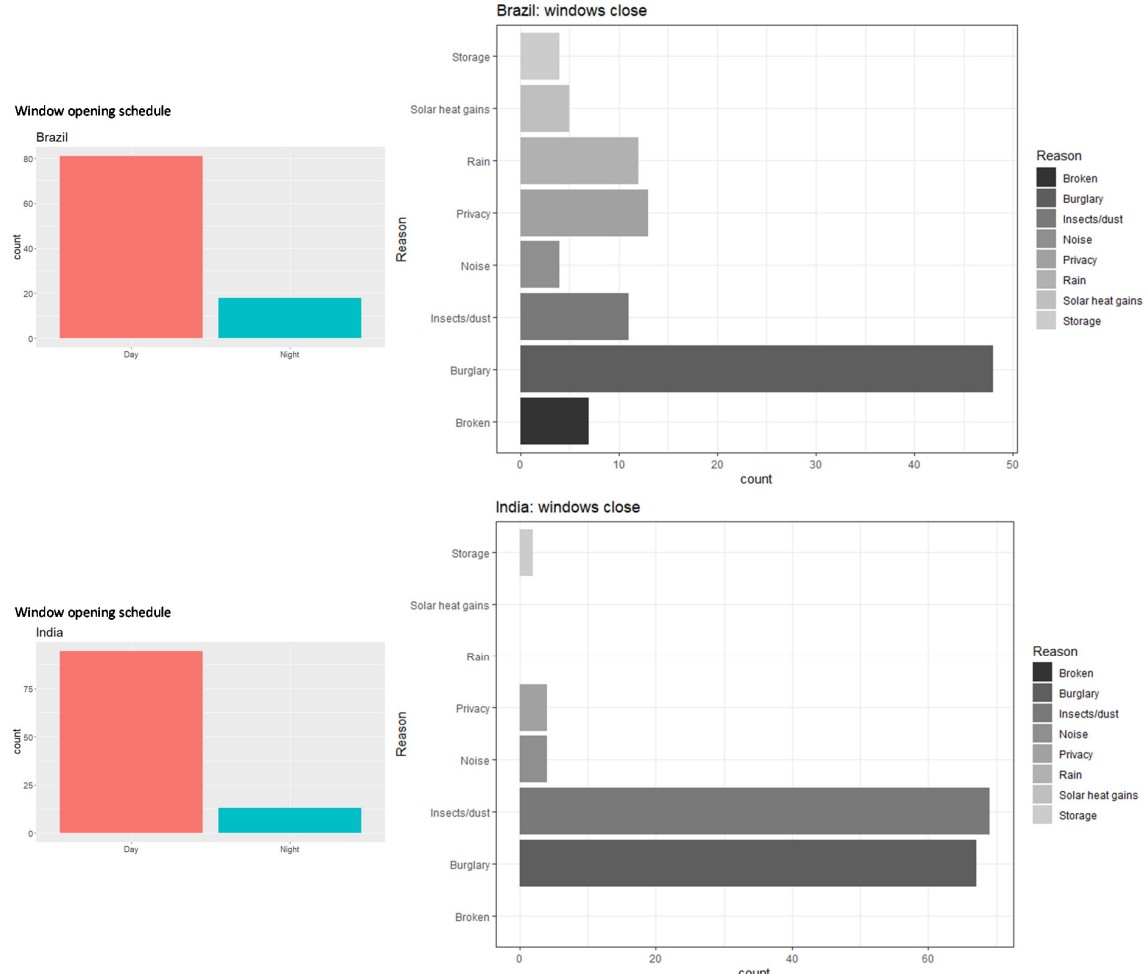

**Figure 7.** Windows opening and closing schedule in the study areas with the reasons for keeping windows closed at night.

*4.2. Empirical Link between Cultural Energy Services and Built Environment Design Elements in the Slum Rehabilitation Housing of Brazil and India*

Firth's bias-reduced binary logistic regression results showed the influence of specific appliance ownership for comfort, cleanliness, and convenience-based energy services. It was found that in both the study areas, the cleaning regime was manual, therefore the absence of energy-intensive cleaning devices (like a vacuum cleaner, see Figure 3). Tables 5 and 6 show the influence of the socio-architectural variables in demand for comfort (Model 1) and convenience (Model 3) appliances in the slum rehabilitation housing (SRH) of Mumbai, India. In Section 4.1, it was observed in Figure 3 that fans were the most common comfort device (97% ownership) in the SRH, Mumbai. High ownership of fans led to a quasi-complete separation problem [68]; hence, the convergence failure of Model 1 with fans as the comfort devices. However, as illustrated in Table 5, rising air conditioner (AC) ownership showed a significant association with a lack of privacy in the study area. It can be inferred that there is a higher likelihood of AC ownership (O.R. = 14.939) in the SRH, Mumbai, due to discomfort due to a lack of privacy. Similar results were reported by [14] as well.

**Table 5.** Firth's bias-reduced regression results for significant energy service demand for comfort in Mumbai, India.

| Lack of Socio-Architectural Elements | Model 1 (Dependent Variable: Appliance Ownership; Yes = 1, No = 0) | | |
|---|---|---|---|
| | Air Conditioners | | |
| | β | Sig. | Exp (β) |
| Privacy | 2.704 | 0.047 * | 14.939 |
| Safety | 1.359 | 0.312 | 3.892 |
| Open space/ventilated space access during night | −2.092 | 0.257 | 0.123 |
| Walkability during day | 1.066 | 0.499 | 2.903 |
| Walkability during night | −0.110 | 0.954 | 0.895 |
| Penalised log likelihood | | −11.865 | |

* $p < 0.05$.

**Table 6.** Firth's bias-reduced regression results for significant energy service demand for convenience in Mumbai, India.

| Lack of Socio-Architectural Elements | Model 3 (Dependent Variable: Appliance Ownership; Yes = 1, No = 0) | | | | | |
|---|---|---|---|---|---|---|
| | Refrigerator | | | Clothing Iron | | |
| | β | Sig. | Exp (β) | β | Sig. | Exp (β) |
| Privacy | −0.895 | 0.358 | 0.408 | −2.320 | 0.080 | 0.098 |
| Safety | −0.659 | 0.142 | 0.517 | 0.460 | 0.325 | 1.584 |
| Open space/ventilated space access during night | −0.857 | 0.381 | 0.424 | 0.515 | 0.600 | 1.673 |
| Walkability during day | 1.649 | 0.049 * | 5.201 | 0.918 | 0.267 | 2.504 |
| Walkability during night | −0.488 | 0.563 | 0.613 | −2.245 | 0.009 ** | 0.105 |
| Penalised log likelihood | | −19.069 | | | −15.998 | |

* $p < 0.05$; ** $p < 0.01$.

The lack of privacy remains a socio-architectural design gap in the surveyed SRH of the Natwar Parekh Complex (also reported in [55]). It was also mentioned in Section 4.1., Figure 7, that one of the main reasons for keeping windows closed at night is due to lack of privacy. It had broader implications on energy demand for comfort at night because household density increases as all the members stay inside. Closed windows and high occupant density (~0.25 person/m$^2$) increase the indoor temperature, and it can explain the rise in the need for energy-intensive cooling demand through AC ownership. Debnath et al. (2019) [20] reported such a change in energy intensity causes high energy bills, which creates a poverty trap [82] for the occupants living in SRH.

Similar significant results were obtained for convenience-related energy demand through the ownership of refrigerators and clothing irons in the Mumbai case study (see Table 6). It can be observed in Table 6 that refrigerator ownership has a higher likelihood (O.R. = 5.201) with lack of walkability in the daytime. It can be associated with the persistent problems of a lack of open and social spaces in the study area, and demand for such convenience-based energy service can be a counter-response to social distress. Such inference remains true to higher clothing iron ownership and lack of walkability at night as well (see Table 6). Comparable results were also reported in [14]. These findings, thus, support our

hypothesis that a lack of socio-architectural elements in the SRH influences the demand for cultural energy services.

    In the SRH of João Pessoa, Brazil, the appliance ownership was observed to be twice that of the SRH in Mumbai (see Figure 3). Therefore, regression results showed a significant influence of fan ownership in the energy service demand for comfort (see Table 7). Results showed that a higher likelihood of multiple fan ownership (O.R. = 5.414) is influenced by the lack of walkability in the daytime. The survey showed that the built environment of the Gadanho and Timbo Social Housing (GTSH) in João Pessoa, Brazil, was relatively better than the SRH in Mumbai in terms of open-space and walkability planning (see Section 3.1). However, it lacked appropriate socio-architectural design compatibility as per the GTSH occupants [59]. Similarly, comfort-specific energy demand was also observed through the higher likelihood of water cooler ownership (O.R. = 18.690) due to lack of walkability at night. Thus, provisioning of walkability in the GTSH can aid in mitigating loss of comfort in the built environment.

**Table 7.** Firth's bias-reduced regression results for significant energy service demand for comfort in João Pessoa, Brazil.

| Lack of Socio-Architectural Elements | Model 1 (Dependent Variable: Appliance Ownership; Yes = 1, No = 0) | | | | | |
| --- | --- | --- | --- | --- | --- | --- |
| | Fan (More Than 1) | | | Water Cooler | | |
| | β | Sig. | Exp (β) | β | Sig. | Exp (β) |
| Privacy | 0.274 | 0.661 | 1.315 | 1.240 | 0.154 | 3.455 |
| Safety | −0.513 | 0.237 | 0.598 | −1.063 | 0.231 | 0.345 |
| Open space/ventilated space access during night | −0.593 | 0.179 | 0.552 | −1.793 | 0.133 | 0.166 |
| Walkability during day | 1.689 | 0.046 * | 5.414 | −0.888 | 0.491 | 0.411 |
| Walkability during night | −1.284 | 0.199 | 0.276 | 2.928 | 0.036 * | 18.690 |
| Penalised log likelihood | | −29.204 | | | −17.683 | |

<div align="center">* $p < 0.05$.</div>

    Table 8 illustrates the regression results of convenience-driven energy services through higher ownership of microwave ovens, washing machines, and ovens. A lower likelihood of microwave oven ownership (O.R. = 0.276) is influenced by poor safety in the GTSH. It is further explained by the negative β-coefficient associated with the "lack of safety" socio-architectural variable (see Table 8). High burglary rates were a substantial built environment problem in the GTSH, as revealed in our surveys (see Figure 7). Similar, the higher likelihood of washing machine ownership (O.R. = 1.373) is influenced by the lack of open and well-ventilated spaces (see Table 8). It indicates the shift in communal washing and drying practices to a more energy-intensive washing regime due to the lack of socio-architectural spaces. Besides, the lack of walkability and small spaces causes inconvenience. It, in turn, influences a higher radio ownership in the GTSH as a counter response (see Table 8).

**Table 8.** Firth's bias-reduced regression results for significant energy service demand for convenience in João Pessoa, Brazil.

| Lack of Socio-Architectural Elements | Model 3 (Dependent Variable: Appliance Ownership; Yes = 1, No = 0) | | | | | | | | |
|---|---|---|---|---|---|---|---|---|---|
| | Microwave Ovens | | | Washing Machines | | | Radio | | |
| | β | Sig. | Exp (β) | β | Sig. | Exp (β) | β | Sig. | Exp (β) |
| Privacy | −0.763 | 0.292 | 0.466 | −0.389 | 0.531 | 0.677 | −1.914 | 0.138 | 0.147 |
| Safety | −1.284 | 0.01 * | 0.276 | −0.257 | 0.554 | 0.773 | −2.148 | 0.106 | 2.166 |
| Open space/ventilated space access during night | −0.676 | 0.179 | 0.508 | 0.317 | 0.047 * | 1.373 | 0.935 | 0.537 | 2.54 |
| Walkability during day | 1.091 | 0.420 | 2.986 | 0.797 | 0.201 | 2.21 | −5.324 | 0.025 * | 0.004 |
| Walkability during night | −1.056 | 0.443 | 0.347 | 0.421 | 0.359 | 1.523 | 1.325 | 0.031 * | 3.758 |
| Penalised log likelihood | −19.069 | | | −29.287 | | | −8.709 | | |

* $p < 0.05$.

Therefore, the regression results presented for Mumbai (see Tables 5 and 6) and João Pessoa (see Tables 7 and 8) showed significant correlations between specific appliance ownership for the 3Cs concerning the lack of socio-architectural variables of slum rehabilitation housing. Understanding such linkages are critical for "good" energy policymaking [22], as it adds a robust planning-driven component to distributive justice.

## 5. Conclusions

This study investigated the energy culture in two typologically distinct slum rehabilitation housings (SRHs) in India and Brazil. The energy cultures in these areas were classified through the demand for specific appliances contributing to comfort, cleanliness, and convenience-driven energy services. The typologically distinct SRH represented the typical layout of such low-income settlements in the hyper-dense cities of the Global South. The SRH case study in Mumbai, India, had a high-rise typology. In contrast, the Brazilian SRH case in João Pessoa had a low-rise building layout. The high-rise typology aimed at maximising occupancy and addressing the housing deficit [14]. The low-rise typology aimed at inclusive design [59]. However, the SRHs under study in Mumbai and João Pessoa had severe socio-architectural design gaps that affected the well-being of the occupants. The effect of socio-architectural incompatibility on demand for comfort, cleanliness, and convenience (3Cs) as cultural energy services was examined. It was assumed that provisioning of the 3Cs in low-income housing along with appropriate built environment design variables could foster distributive energy justice.

An empirical model was developed using Firth's binary logistic regression to reduce small-sample bias. The model evaluated the ownership of specific 3C appliances concerning the lack of certain socio-architectural variables. It was found that the lack of open spaces and walkability in both the study areas may have influenced the higher demand for comfort- and convenience-specific energy services as a rebound response. Therefore, integrating socio-architectural design elements in slum rehabilitation planning can foster distributive energy justice through appropriate 3C provisioning. The key conclusions that can be drawn from this study are:

- The typology of the slum rehabilitation built environment can act as a critical control variable for distributive energy justice planning, as it influences the demand for cultural energy services and specific appliance ownership.
- An appropriate socio-architectural design of the slum rehabilitation housing can support the local social networks through access to open spaces and well-ventilated areas. These variables are crucial for occupants' demand for comfort, cleanliness, and convenience (3Cs) as energy services. Higher demand for the 3Cs can foster better eudemonic well-being in low-income urban populations (after [13]). Thus, translating the welfare effects of the 3Cs as distributive justice.

■　The empirical model showed that cultural energy services were demanded as a counter-response to the lack of appropriate socio-architectural design variables in the slum rehabilitation housing (SRH) of Mumbai, India, and João Pessoa, Brazil. However, the effects were different due to distinct building typologies. The SRH in Mumbai had a high-rise built form that had no provisioning of social and open spaces. Lack of privacy and safety was the main reason behind keeping windows closed for the most part of the day. It caused higher discomfort, leading to a rise in air conditioner ownership that increased the energy intensity of the households.

■　The SRH in Brazil had a low-rise built form with relatively better access to open spaces and walkable areas. However, it could not satisfy the socio-architectural needs of the occupants. The empirical model showed that most convenience appliances were owned due to this socio-architectural incompatibility. For example, a higher likelihood of washing machine ownership in the study area is linked to poor access to open and ventilated spaces. Such social spaces were used for communal washing and drying regimes, which was lost during slum rehabilitation planning.

■　Lack of privacy and safety were common concerns in both the SRHs, which also showed to have a higher likelihood of demand for convenience specific energy services. In Mumbai, it translated into higher AC ownership; thus, unaffordability of energy bills. Whereas, in Brazil, lack of such socio-architectural variables translated into a higher likelihood of microwave oven ownership, an energy-intensive convenience device.

The policy implications of this study address the multiple disciplinary concerns of urban planning, energy sustainability, and poverty alleviation. Distributive energy justice policies for slum households must include the socio-architectural built environmental needs like open spaces, higher privacy gradients, as well as better safety, sanitation, and hygiene. It links sustainable energy provisioning in resource-constrained settings with built environment planning for ever-increasing low-income populations in the rapidly urbanising Global South; thus, contributing to the contemporary discussions on "good" energy policy [22]. A built environment inclusive energy planning can aid in a better tariff mechanism for the low-income population. Such that energy access does not become a poverty trap [82,83]. Besides, such design-led interventions can further strengthen the policy impact of slum rehabilitation programs regarding the UN's Sustainable Development Goals, specifically, SDG 7— (clean and affordable energy)—and SDG 11— (sustainable societies and communities).

While this study established a bias-reduced empirical model using a robust small-sample regression technique, the generalisability of the model remains a limitation of this study. The limitation is also due to high heterogeneity in the slum rehabilitation contexts across the Global South. Understanding the granular details about the socio-cultural logic of energy demand in poverty can aid in better energy provisioning in such low-income communities. It can further enhance the welfare effects of social and energy policies. Therefore, contributing to the current discussions on distributive justice for poverty alleviation. Our future work will focus on improving the robustness and scope of the preliminary empirical model by integrating more slum rehabilitation building typologies from the Global South. It will create a database of energy cultures across different socio-architectural contexts of slum rehabilitation housing. It can aid planners and policymakers in evidence-driven decision making.

**Supplementary Materials:** The following are available online at http://www.mdpi.com/2071-1050/12/7/3027/s1, S1: Survey questionnaires.

**Author Contributions:** Conceptualization, R.D.; methodology, R.D.; software, R.D.; visualization, R.D.; formal analysis, R.D.; investigation, R.D., G.M.F.S. and R.B.; resources, R.D.; data curation, R.D., G.M.F.S., R.B. and S.M.L.; writing—Original draft, R.D.; writing—Review and editing, R.D., R.B., S.M.L., R.L. and M.S.-B.; supervision, M.S.-B.; project administration, R.D. and S.M.L.; funding acquisition, R.D. and R.B. All authors have read and agreed to the published version of the manuscript.

**Funding:** Part of this research was funded by Bill and Melinda Gates Foundation, through the Gates Cambridge Scholarship under the grant number [OPP1144], Trinity College MCSC Scholarship (Honorary) and the Santander Mobility Grant awarded to (R.D.). Part of this research was also funded by the Ministry of Human Resource Development, the Government of India (GoI) project titled CoE-FAST under the grant number [14MHRD005] and

the IRCC-IIT Bombay Fund, under the grant number [16IRCC561015] awarded to (R.B.). The APC was funded by Bill and Melinda Gates Foundation.

**Acknowledgments:** The authors acknowledge the residents of Natwar Parekh Complex in Mumbai, India, and Gadanho and Timbo Social Housing in Joao Pessoa, Brazil for their continuous support and patience.

**Conflicts of Interest:** The authors declare no conflict of interest. All opinion, findings and conclusions or recommendations presented in the paper are that of the authors and it do not necessarily reflect the views of the associated organisations.

## Appendix A

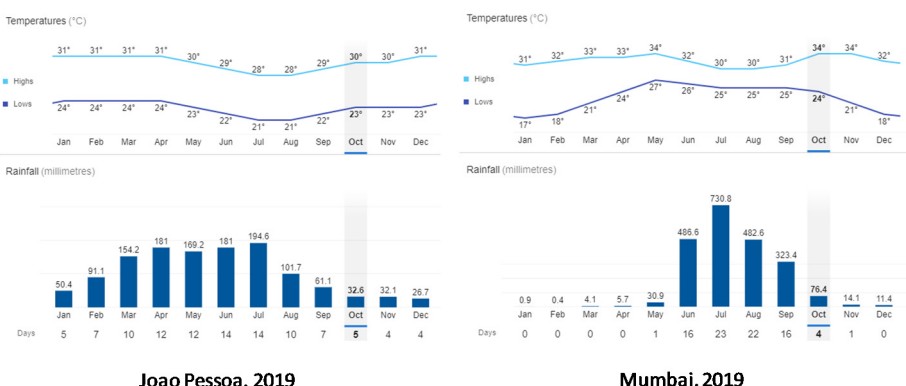

**Figure A1.** Climatic variation of Joao Pessoa, Brazil, and Mumbai, India, for 2019 (source: Google weather).

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
