# Peer review of "Energy Justice in Slum Rehabilitation Housing: An Empirical Exploration of Built Environment Effects on Socio-Cultural Energy Demand"

_sustainability, doi:10.3390/su12073027_

Round 1

Reviewer 1 Report

Dear authors

The article

“Energy justice in slum rehabilitation housing: An empirical exploration of built environment effects on socio-cultural energy demand”

explores energy justice, which is an important topic being studied across many academic disciplines in energy research. I commend the authors for the choice of topic, data collection, and the amount of work performed.  There are many errors in the manuscript, and the writing lacks focus and quality. Therefore, my suggestion is revising the entire manuscript for correctness, clarity, focus, and conciseness seeking assistance from an experienced author in the field. Further, the authors should seek assistance from a person whose English proficiency is close to that of native speakers.

The following includes specific points that I find noteworthy in the case of this manuscript. First, please pay more attention to the way key concepts are defined in the text. For example, consider line 38 “Energy justice has emerged as a critical instrument of good energy policy” and line 76: “Slum rehabilitation is a policy instrument of slum upgradation in rapidly urbanizing economics”. Energy justice and slum rehabilitation are not instruments, and economics is a branch of knowledge. The authors should insert a cited definition where such key concepts appear for the first time in the text (see, e.g., LaBelle, 2017). Second, please minimize the potential for uncertainty and ambiguity. For example, consider the use of “it” at the beginning of the sentences of lines 47 and 57: “It translates everyday …” “It is collectively called …” It is not clear to the reader what the “It” refers to in these examples. Third, please avoid the use of unnecessarily long sentences and complex terms intended to convey simple ideas. Good manuscripts are often written in the most concise and simple way possible, thus ensuring that they are focused and easy to understand. When re-writing the manuscript, please follow the journal’s instructions for authors (https://www.mdpi.com/journal/buildings/instructions).

Author Response

We thank the reviewer for the comments. The manuscript has now been carefully edited to address the reviewer's comment. Please see the track changed version of the manuscript. The writing is improved for correctness, clarity, focus, and conciseness. 

Specifically, 

line 38 'Energy justice has emerged as a critical instrument of good energy policy...' has been amended to lin3 38-41:

'... An energy-just world is believed to promote happiness, welfare, freedom, equity and due process for both producers and consumers [1] (pp 13). Energy justice is a critical element of contemporary energy policies addressing climate change mitigation and sustainable development goals....' 

line 76 '... Slum rehabilitation is a policy instrument of slum upgradation in rapidly urbanizing economics...' 

We thank the reviewer for this comment, we have now corrected it as '... Slum rehabilitation programs are slum upgradation policies that enable slum dwellers to own a formal house that can improve their quality of life [15]...' (line 79)

The authors should insert a cited definition where such key concepts appear for the first time in the text (see, e.g., LaBelle, 2017). Second, please minimize the potential for uncertainty and ambiguity. For example, consider the use of “it” at the beginning of the sentences of lines 47 and 57: “It translates everyday …” “It is collectively called …” It is not clear to the reader what the “It” refers to in these examples.

We thank the reviewer for this comment, we have now amended it to: 

'... Energy cultures are derived from everyday energy practices, norms and the material reality of the built environment that drives the need for specific energy services [6]. Energy culture translates everyday energy consumption into household welfare which promotes energy justice. It is the responsibility of an energy just system to increase welfare by improving individuals’ capabilities for maximising utility [7]....' (line 47-51)

Reviewer 2 Report

This paper explores slum rehabilitation and energy justice in Brazil and India. The paper presents a detailed background and appropriate methods investigate the research questions, through surveys and bivariate and multivariate analyses. The paper need a few minor edits that I will detail below. 

  1. Double-Check the references numbers, for instance 53 and 54 are a repeat of the same reference.
  2. The image quality of the figures made it difficult to read.

Author Response

We thank the reviewer for the comments. We have amended the manuscript as per the comments. 

1. Double-Check the references numbers, for instance, 53 and 54 are a repeat of the same reference.

Thank you for pointing this out. We have now double-checked the references and omitted the duplicate entries. 

2. The image quality of the figures made it difficult to read.

Thank you for this comment. We have now re-iterated some of the figures in the word document. However, we think the journal downscales the image in the pdf format. Please check the image quality in the revised word file.

Reviewer 3 Report

The authors have produced some very interesting findings taking the theoretical framework of Shove and Sovacool applying it to built environmental implications for social housing in Brazil and India. The concepts of energy justice and energy culture is very crucial when analysing or even implementing sustainable low income urban housing in the rapid urbanising contexts of global south. The section on materials and methods clearly set out the research design and the contextual differences is clearly set out for the reader to provide the basis for the nuanced analysis in section 4. The Discussion indicating the critical analysis and conclusion are well written based on the evidence and literature previously discussed in the paper. 

Overall, an excellent paper.

Author Response

We express our sincere gratitude to the reviewer for the comments. Now we have edited the manuscript to improve the readability and consistency. We have also carefully checked the manuscript for spelling mistakes.

Round 2

Reviewer 1 Report

For the authors

The manuscript is not acceptable as a journal paper. There are many errors in the manuscript, and it still lacks focus, organization, quality, and rigor of a scientific paper. You should have provided one clean version and one tracked version of the original manuscript. I only received the tracked version, which is hard to read and unprofessional. The following includes specific points intended to help increase the quality of the manuscript:

Line 19: “Understanding these drivers become …” > “… becomes …”

Line 20: “… more necessary in poverty …” > “… in impoverished areas …”

Line 20: “… leading to energy injustices.” > I wouldn’t include this here. The fact that low-income occupants struggle to avail essential energy services does not lead to injustice by itself.

Line 21: “Literature indicates that such injustices can …” > I would change this to “… that energy injustice can …”

Line 32: “… through appropriate provisioning of socio-architectural and 3Cs guided energy demand in poverty.” > Since energy demand is not provided, I would change this to “… through appropriate planning for …”  

Line 41: “Much of the energy-justice based-application framework is designed for investigating and restructuring the supply-side of energy systems for delivery equity in the society and individual’s life.” > I would change this to “Energy justice frameworks have been designed to investigate and restructure the supply of energy and enhance equity.” 

Line 44: “that determines” > “that determine”

Line 55: “services which enable them to their eudemonic well-being” > maybe “services which enhance their eudemonic well-being”?

Line 55: “However, the thresholds for minimum energy services are not yet fully understood. The current literature …” The first sentence is redundant. I would change it to “However, the current literature …”

Line 56  “The current literature lacks evidence on the thresholds of a minimum of energy services as energy-consumption at a household-level is principally viewed as a physical quantity.” I think you may refer to, or at least cite, the following paper here in your paper:

Walker, G., Simcock, N., & Day, R. (2016). Necessary energy uses and a minimum standard of living in the United Kingdom: Energy justice or escalating expectations?. Energy Research & Social Science, 18, 129-138.

Line 57: “… as energy-consumption at a household-level …” > “… as energy consumption at the household level …” is correct.

Line 60: “… at a human-level …” > I would use “at the individual level”

Line 69: “… hyper-modern appliances that save ‘time’. For example: storing cooked food in the refrigerator rather than cooking every meal.” > From the writing standpoint, this sentence is not professional. I would change it to “… hypermodern time-saving appliances (e.g., storing cooked food in the refrigerator rather than cooking every meal)”. Is a refrigerator a hypermodern appliance? I don’t think so. You put refrigerators under welfare appliances in Table 2. 

Line 73: “… poorer-household demand energy services for ‘subsistence’ ...” > “… poorer households demand energy services for subsistence ...” Hyphen is used to join two or more words serving as a single adjective before a noun. (see, e.g., https://owl.purdue.edu/owl/general_writing/punctuation/hyphen_use.html) If you are writing in American English, use double quotation marks at all times unless quoting something within a quotation, when you use single. Also, there is no reason to put single words in quotation marks, unless you want to emphasize a word someone else used. Usually, this implies that the author doesn't agree with the use of the term (which does not apply here). Therefore, I would not put single words in quotation marks.

Line 78: What is “fulling aspirations”?

Line 81: “Slum rehabilitation programs are slum upgradation policies …” I do not agree with this sentence. Programs are different from policies, but the distinction is disappeared here. A policy can be defined as “A relatively stable, purposeful course of action or inaction followed by an actor or set of actors in dealing with a problem of concern.” (Anderson, J.E. (1975) Public Policymaking. Praeger). There is often a hierarchical order of planning process in the form of policy-program-project-procedure, meaning that policies encompass programs, programs encompass projects, and projects encompass procedures. Perhaps more importantly, in Line 83 you start by “Slum rehabilitation policies …” which is confusing when compared to “Slum rehabilitation programs …” in Line 81.  

Further, I would remove these two sentences in Lines 81-82: “Slum rehabilitation programs are slum upgradation policies that enable slum dwellers to own a formal house that can improve their quality of life. Social housing also falls under this category [16]”. Instead, I would rewrite Line 83 as “Slum rehabilitation aims at improving the quality of life and eudemonic well-being of the urban poor by enabling slum dwellers to own a house”.

There are introductory books about policymaking. See, e.g., Birkland, T. A. (2019). An introduction to the policy process: Theories, concepts, and models of public policymaking. Routledge.

Line 84: “However, the inferior quality of rehabilitation built environment planning showed counterintuitive results towards energy sustainability [13], health and well-being sustainability [17], [18] and loss of social network [19]” This sentence is confusing because 1) you switch to the past tense, whereas previous and next sentences are written in the present tense 2) you introduce the word planning, whereas the previous sentences are about policies and programs 3) The concepts energy sustainability and health and well-being sustainability are positive, whereas loss of social network is negative. Mixing these concepts is incorrect from the writing standpoint. Therefore, I would rewrite the sentence “However, the inferior quality of rehabilitation built environment planning showed counterintuitive results towards energy sustainability [13], health and well-being sustainability [17], [18] and loss of social network [19]” as “However, low-quality slum rehabilitation can negatively impact energy sustainability and health, well-being, and socialization of the urban poor [13-19].”

Line 86-88: “In the same study, authors had discussed that the lack of open spaces had disrupted the social network of the occupants.” Again, this is not consistent with its previous sentence. I would rewrite as “A recent study on slum rehabilitation housing in India shows that low-quality built environment pushes occupants towards energy poverty by increasing their household energy bills [20]. In the same study, the authors discuss that the lack of open spaces has disrupted the social network of the occupants. We argue that …”

Line 89-93: “We argue that the poor design of slum rehabilitation built environment is a distributive injustice that is restricting the welfare benefits of cultural energy services (3Cs) in the study areas. Therefore, the influence of slum rehabilitation built environment is investigated in the delivery of comfort, cleanliness and convenience in poverty through appliance ownership.” I would rewrite this as “We argue that low-quality slum rehabilitation can disproportionately restrict the welfare benefits of cultural energy services (3Cs) among poor households. Therefore, we investigate the influence of slum rehabilitation on the delivery of comfort, cleanliness and convenience through appliance ownership.”    

Line 94: “We vary two variables in this study.” This part should be placed after the description of the research objectives, which you put in Line 158-166. By the way, it is not common to describe study variables in the introduction section. These variables are only described in the methodology section.

Line 98: “… and convenience (3Cs) poverty …” what is poverty doing here?

Line 149: “… socio-technical inference of architecture …” By inference, do you mean understanding?

Line 179: “Brand-Correa et al. (2018)[11] have explored …. They found …” In a scientific paper, this is considered a tense consistency problem.

Line 183: Several times throughout this section (Background/Literature Review) you mention your contributions, opinions, etc. For example, “our study investigates …” in Line 183, “Here, we study appliance uptake …” in Line 270, “We find that the above studies …” in Line 285, “This study expands this …” in Line 293, etc. This is not how scientific papers are typically written. You should keep these for the end of the Background/Literature Review section. Otherwise, you are adding complexity and confusion.

Line 185: “… in low-income.” This is grammatically incorrect. This should change to “… among low-income households.” or something similar.

Line 186: “… definitions of 3Cs by Shove, (2003)[10] …” and “… energy service ladder concept by [12] …” These two references should be cited consistently.

Line 200: I skipped this table and the text in the Background section, but it requires significant English editing.

Line 379: Times and dates of data collection are missing.

 Line 381: You should have included the survey questionnaire in the Appendix for the reviewers. In the current form, the study does not satisfy the replicability criterion.

Line 386: “We chose India and Brazil to support the control variables of this study (see section 1), i.e., variation in SRH building typology and socio-cultural factors of 3Cs.” This sentence is redundant.

Line 390: “Natwar Parekh Compound (NPC) …” If this describes the study area, it should come in the next section (Study area characteristics) not in the Data collection and survey design section. The study area is typically introduced before introducing the data and variables.

Line 664-684: “Reports claim ... However, studies have shown … However, existing literature also shows …” Either provide citations or remove these types of sentences.

Line 769: “A binary logistic regression modelling …” Why do you put your model description under the title João Pessoa, Brazil: Gadanho and Timbó Social Housing (GTSH)?

Line 829: “Higher TV ownership indicates a stronger demand for convenience-based energy service in Brazilian households.” To make such statements, you need to convince the reader that the availability and cost of ownership of such appliances in India are at least like - if not the same as – those in Brazil. Even in that case, there are still other factors that are out of your control. For example, Table 3 shows that Brazilian households have higher incomes and benefit from low-income electricity tariff programs. The same applies to Line 842 “Besides, appliances like blenders, DVD players, coffee machines, juicer, microwave ovens, …” Maybe these appliances are just less expensive for Brazilian households. If these are just cheaper for Brazilians, then you may not attribute the consumption of such products to cultural differences. What if there is a local home appliance manufacturer in the Brazilian case that produces TVs that are sold to local households for up to 50% off their exported prices because of the tariff program? I think you can present such descriptive data, but you cannot make firm conclusions at this point.

Line 882: In this and other figures, the authors should better present both areas in one chart to reduce page number and make the comparison easier.  

Line 893: “This level of occupancy is due to the strong cultural norm that at least one member (mostly women) of the family stays at home to take care of children and grandparents” These kinds of statements should be supported by data – here on children and grandparents, which are missing.  

Line 917: If all the data in these charts can be described in two sentences, you should better save pages.

Author Response

Response to reviewer 1

The manuscript is not acceptable as a journal paper. There are many errors in the manuscript, and it still lacks focus, organization, quality, and rigor of a scientific paper. You should have provided one clean version and one tracked version of the original manuscript. I only received the tracked version, which is hard to read and unprofessional.

We thank the reviewer for the detailed comments in this round of reviews. We have worked hard on the suggestions and made necessary amendments to improve the quality of the paper.

At this second revision stage, we would like to point out that we respectfully disagree to reviewer 1’s harsh comments ‘…. The manuscript is not acceptable as a journal paper. There are many errors in the manuscript, and it still lacks focus, organization, quality, and rigor of a scientific paper….’. This is because, we have received commendable comment on this manuscript from reviewer 2 and reviewer 3. It was to the extent that the other two expert reviewers suggested minor revisions with language editing. Their comments are appended for your reference over here:

“Reviewer 2: This paper explores slum rehabilitation and energy justice in Brazil and India. The paper presents a detailed background and appropriate methods investigate the research questions, through surveys and bivariate and multivariate analyses. The paper needs a few minor edits.

Reviewer 3: The authors have produced some very interesting findings taking the theoretical framework of Shove and Sovacool applying it to built environmental implications for social housing in Brazil and India. The concepts of energy justice and energy culture is very crucial when analysing or even implementing sustainable low-income urban housing in the rapid urbanising contexts of global south. The section on materials and methods clearly set out the research design and the contextual differences is clearly set out for the reader to provide the basis for the nuanced analysis in section 4. The Discussion indicating the critical analysis and conclusion are well written based on the evidence and literature previously discussed in the paper. Overall, an excellent paper.”

To answer the comment ‘…You should have provided one clean version and one tracked version of the original manuscript. I only received the tracked version, which is hard to read and unprofessional….’

The journal office asked us to upload the track changed manuscript and a pdf version of the revised manuscript. It is out of our control to determine what version is sent to you. We strongly believe that an experienced expert reviewer must be well-aware of it.

The point-by-point responses are appended in the attached document

Round 3

Reviewer 1 Report

Sustainability                          

I am writing regarding the article

“Energy justice in slum rehabilitation housing: An empirical exploration of built environment effects on

socio-cultural energy demand”

I appreciate the authors for providing responses and incorporating my comments in the text. However, I cannot accept this manuscript for Sustainability because there are still serious errors in the text and issues left unanswered.

First, I had asked the authors to include their questionnaire as a supplementary file. The authors have responded that they cannot share the questionnaire at this stage due to ethical concerns. I cannot think of any ethical considerations in sharing the questionnaire with the journal during the peer review process. The language of a questionnaire can have a significant impact on the way the respondents answer the questions. Without a reliable questionnaire, the study results are not reliable and the study is not replicable.

Second, the authors have incorporated my comments regarding writing errors in the text up to a certain point. The rest of the text, like the Results and Discussion and Conclusion sections, have remained untouched. Therefore, there are still many errors in the text. The authors should have taken this matter seriously because it is not the responsibility of the reviewer to correct all the writing errors every time the manuscript is circulated. Rather, the authors should have acknowledged the fact that the text needed extensive English editing.    

Third, the statistical models – therefore the conclusions - are not reliable. I have not seen this way of presenting a regression model in more than 100 regression-based papers I have read. It is very uncommon to mix variables, coefficients, and descriptions when presenting statistical models. Moreover, none of the coefficients are statistically significant (at p<0.05 level) and the model fit summary (e.g., using Nagelkerke R Square) suggests that the models have not resulted in even an average fit. Besides, standard errors and average odds ratios are missing. Another factor that says this model is not reliable is the differences between lower- and upper-bound odds ratios, which are extremely high. In other words, the amount of variability is too high. When statistical models lack statistically significant coefficients, models do not result in good fits, and variabilities are extremely high, the models cannot be relied upon to reach any conclusions. 

Author Response

Response to reviewers:

First, I had asked the authors to include their questionnaire as a supplementary file. The authors have responded that they cannot share the questionnaire at this stage due to ethical concerns. I cannot think of any ethical considerations in sharing the questionnaire with the journal during the peer review process. The language of a questionnaire can have a significant impact on the way the respondents answer the questions. Without a reliable questionnaire, the study results are not reliable and the study is not replicable.

Response:

We bestow our heartiest gratitude to the reviewer for painstakingly reviewing our manuscript for the third round. We are indebted to the reviewer for the comments that are improving the quality of this manuscript by leap and bounds. We apologise for our earlier response of not sharing the questionnaire for ethical reasons. We misunderstood that the questionnaire will be online with the paper as this research is a part of a wider project which has strict ethical guidelines. However, for the peer-review process, we are uploading the questionnaire now as supplementary material.

Once again, we are grateful to the reviewer for your time and energy on providing such much-needed high-quality comments.

Second, the authors have incorporated my comments regarding writing errors in the text up to a certain point. The rest of the text, like the Results and Discussion and Conclusion sections, have remained untouched. Therefore, there are still many errors in the text. The authors should have taken this matter seriously because it is not the responsibility of the reviewer to correct all the writing errors every time the manuscript is circulated. Rather, the authors should have acknowledged the fact that the text needed extensive English editing.   

Response:

We thank the reviewer for the detailed editorial comments on the earlier version of the manuscript. We do not agree on this point that we have left the text untouched especially in the Results and Discussion section. We could not include the point by point edits in the last response to reviews as the file was getting too large, however, we did humbly request the reviewer to crosscheck the tracked-change file.

We can assure you that we take your experts comments very seriously and have worked hard to amend the mistakes that you have pointed out, both structurally and editorially. Even in this version, we have made significant edits to the Results and Conclusion section. However, we do acknowledge your suggestion that further English editing may be required. 

In doing so, we have completely rewritten the section 4.2 and the Conclusion section. We hope that we have addressed your comments satisfactorily. Nonetheless, we remain indebted to your dedication and professionalism.

Third, the statistical models – therefore the conclusions - are not reliable. I have not seen this way of presenting a regression model in more than 100 regression-based papers I have read. It is very uncommon to mix variables, coefficients, and descriptions when presenting statistical models. Moreover, none of the coefficients is statistically significant (at p<0.05 level) and the model fit summary (e.g., using Nagelkerke R Square) suggests that the models have not resulted in even an average fit. Besides, standard errors and average odds ratios are missing. Another factor that says this model is not reliable is the differences between lower- and upper-bound odds ratios, which are extremely high. In other words, the amount of variability is too high. When statistical models lack statistically significant coefficients, models do not result in good fits, and variabilities are extremely high, the models cannot be relied upon to reach any conclusions.

Response:

We are particularly indebted to the reviewer for pointing out this modelling error. However, before we respond to this query, we would like to clarify why we did not fine-tune the model. The idea behind the analysis was not to perform econometric modelling of the variables but rather test the directionality of the energy service variables with the built environment variables by analysing the signs of beta-coefficient. It shifted our focus from the numbers, and we devoted our energy on interpreting the signs of beta-coefficients. We found a similar approach to many small-sample regression studies. We did acknowledge that the lack of generalisability of this model remains a limitation of this study, but we will be validated with a larger dataset from other slum rehabilitation cases as well as our future work.

We found that the problem with the Nagelkerke R Square not reporting as per the average fit is due to separation problem with the dataset. Some appliances like Fan, TV and refrigerator have high ownership in both the cases, that created a problem of quasi-complete separation. Allison, 2008 provided a very concise explanation to this problem (http://www.people.vcu.edu/~dbandyop/BIOS625/Convergence_Logistic.pdf).

Having said that, we completely agree with your comments and we are grateful that you have been strict about it. We dug deeper into the current literature and found that Firth’s binary logistic regression approach for reducing bias using penalised log-likelihood represents the state-of-the-art in the small-sample analysis. We reran our model and found this model improved the results, and to save further confusion we have only reported the significant results. Please see the revised methodology section 3.3 and the improved results in section 4.2. We are also attaching those sections as a response to this review, please find in the attached file. We are sincerely grateful to you for this invaluable comment!

Furthermore, thank you for commenting on the presentation of our regression results. Now we have revised it as per the standard APA table presentation guidelines.

Thank you once again for such valuable comments and we hope that our paper is now up to the expected standard. We have re-edited the paper to reduce the language errors, however, it might need further editing.
